# CondiDiag1.0: A flexible online diagnostic tool for conditional sampling and budget analysis in the E3SM atmosphere model (EAM)

Hui Wan[1], Kai Zhang[1], Philip J. Rasch[1], Vincent E. Larson[1,2], Xubin Zeng[3], Shixuan Zhang[1], and Ross Dixon[4]

[1]Atmospheric Sciences and Global Change Division, Pacific Northwest National Laboratory, Richland, Washington, USA
[2]Department of Mathematical Sciences, University of Wisconsin–Milwaukee, Milwaukee, Wisconsin, USA
[3]Department of Hydrology and Atmospheric Sciences, University of Arizona, Tucson, Arizona, USA
[4]Department of Earth and Atmospheric Sciences, University of Nebraska–Lincoln, Lincoln, Nebraska, USA

**Correspondence:** Hui Wan (Hui.Wan@pnnl.gov)

**Abstract.** Numerical models used in weather and climate prediction take into account a comprehensive set of atmospheric processes (i.e., phenomena) such as the resolved and unresolved fluid dynamics, radiative transfer, cloud and aerosol life cycles, and mass or energy exchanges with the Earth's surface. In order to identify model deficiencies and improve predictive skills, it is important to obtain process-level understanding of the interactions between different processes. Conditional sampling and budget analysis are powerful tools for process-oriented model evaluation, but they often require tedious ad hoc coding and large amounts of instantaneous model output, resulting in inefficient use of human and computing resources. This paper presents an online diagnostic tool that addresses this challenge by monitoring model variables in a generic manner as they evolve within the time integration cycle.

The tool is convenient to use. It allows users to select sampling conditions and specify monitored variables at run time. Both the evolving values of the model variables and their increments caused by different atmospheric processes can be monitored and archived. Online calculation of vertical integrals is also supported. Multiple sampling conditions can be monitored in a single simulation in combination with unconditional sampling. The paper explains in detail the design and implementation of the tool in the Energy Exascale Earth System Model (E3SM) version 1. The usage is demonstrated through three examples: a global budget analysis of dust aerosol mass concentration, a composite analysis of sea salt emission and its dependency on surface wind speed, and a conditionally sampled relative humidity budget. The tool is expected to be easily portable to closely related atmospheric models that use the same or similar data structures and time integration methods.

## 1 Introduction

Atmospheric general circulation models (AGCMs) used in climate research and weather prediction are simplified mathematical representations of the complex physical and chemical processes (phenomena) driving the evolution of the Earth's atmosphere. Despite the necessity of simplification due to the limitation in computing resources, it is highly desirable that, to the extent possible and practical, models should be based on first principles and robust quantitative relationships in atmospheric physics

and chemistry, so that the same models can reliably provide good accuracy under historically observed atmospheric conditions as well as in the climate of the future. Many tools have been used for assessing the behavior and fidelity of the atmospheric processes represented in numerical models. Among those, budget analyses are a useful method for quantifying relationships between different processes, and composite analyses are useful for revealing the characteristics of atmospheric conditions and their changes under specific situations. Both methods have been widely used in process-oriented model evaluation to help identify model deficiencies and improve predictive skills. Carrying out such analyses, however, often requires tedious ad hoc coding.

Consider, for example, a model evaluation study aiming at understanding the role of various processes in influencing the simulated atmospheric water cycle, which involves specific humidity $q_v$ as a prognostic variable of the AGCM. The typical way to obtain a budget of $q_v$ is to review the model source code, manually add extra lines of code and variables into subroutines representing parameterizations and the dynamical core to save the rate of change (i.e., tendency) of $q_v$ caused by each process of interest, and then archive those tendencies in model output. Since modern AGCMs are sophisticated, a complete budget analysis with the finest granularity will likely involve a number of tendency terms. If a researcher wishes to obtain several different views of the $q_v$ budget with different levels of granularity (e.g., considering all stratiform cloud processes as a single $q_v$ tendency term in one budget but breaking it down to evaluating condensation/evaporation and rain-formation processes separately in a second view), then the tendencies of coarser granularity will either need to be computed from the fine-grained terms during post-processing or be calculated online and saved in additional model variables. Modern AGCMs often include multiple water species as prognostic variables and tens to hundreds more variables representing aerosol and gas species. Some models also include diagnostic variables such as isotopes and tagged water or aerosol species originating from different geographical regions (e.g., Wang et al., 2014; Zhang et al., 2015; Singh et al., 2016; Bailey et al., 2019; Wang et al., 2020). The lines of code and additional variables that are needed to monitor, assess, and diagnose tendency terms can quickly add up to a huge number, increasing code complexity, computational overhead, and the potential for bugs both in the code and during post-processing.

An AGCM also often contains many diagnostic variables that are needed in the equations of a parameterization. For example, the relative humidity with respect to ice (RHI) is often used in the prediction of formation of cloud ice crystals (cf. Sect. 6.3). While an AGCM might only calculate RHI once or a few times during each time step, a detailed budget analysis of RHI tracking all mechanisms affecting the air temperature, pressure, and specific humidity can provide useful insights into the atmospheric processes that contribute to or compete with ice cloud formation. These types of diagnostic variables appear frequently in AGCMs, and supporting budget analyses for them would require inserting many new model variables and output, which often leads to a dilemma in source code management: that if a user throws away the ad hoc code after their study is completed, other users interested in similar topics will need to reinvent the wheel or at least re-do the coding; on the other hand, if users commit study-specific code to the model's central repository, clutter will accumulate quickly.

Similar challenges are encountered in studies involving composite analysis, the essence of which is to define a criterion, conditionally sample some model variables, and then analyze the stratified data to look for relationships occurring under the specific condition. Conditional sampling in AGCM simulations is often carried out by first archiving a large amount of instantaneous model fields at a sufficiently high frequency and then using post-processing to produce the conditionally sampled

composite (see, e.g., Ghan et al., 2016; Gryspeerdt et al., 2020). This not only can lead to inefficient use of computing time (due to I/O bottleneck) but also creates challenges in data storage and transfer. Occasionally, conditional sampling is carried out online (i.e., during a simulation) so that only the temporal averages of model variables meeting the sampling condition need to be archived. With this approach, ad hoc coding is often used for each combination of sampling condition and monitored variable, which again results in challenges in code management.

Authors of the present paper recently started an effort to identify and address numerical artifacts in the time integration methods used by physics parameterizations and process coupling in version 1 of the atmosphere component of the Energy Earth System Model (EAMv1, Rasch et al., 2019; Xie et al., 2018; Golaz et al., 2019). The study of Wan et al. (2021) and its follow-up investigations have involved monitoring not only EAM's prognostic variables but also non-standard output fields such as various measures of supersaturation and atmospheric instability. Those investigations constantly require the use of composite and budget analyses, motivating our development of a new, general, and user-friendly online diagnostic tool to facilitate the investigations. This paper presents the first version of the new tool, which we refer to as CondiDiag1.0.

Assuming the physical quantities to be monitored already exist in EAM, configuring a simulation to activate CondiDiag will normally require only setting a small number of switches in the model's input file (currently using Fortran namelist, cf. Sect. 5.2). A minimal amount of special-purpose code might be required from the user if existing model variables need to be monitored at new locations in the model's time loop, if the variables exist within a parameterization or the dynamical core but need to be made available in the data structures accessible by our tool, or if a quantity of interest is not available in the original EAM and needs to be calculated from the existing variables. The coding required in such cases will be relatively simple. To facilitate budget analyses, the tool provides the flexibility to monitor and archive both the evolving values of model variables and their increments caused by different atmospheric processes. Vertical integrals are calculated online when they are requested through namelist. Multiple sampling conditions can be used in the same simulation. Unconditional sampling and mixtures of conditional and unconditional sampling are also supported.

The new tool was designed for and implemented in EAMv1. It has been ported to EAMv2 and also to a few code versions in between. We expect it to be straightforward to port the tool to EAMv1's recent predecessors, e.g., the Community Atmosphere Model versions 5 and 4 (CAM5 and CAM4, Neale et al., 2012, 2010), as well as their other descendants (e.g., CAM6, Craig et al., 2021), as these models use the same Fortran derived data types for organizing information passed through the physics parameterizations suite. Examples of such Fortran data types include the "physics state", "physics buffer", atmosphere "import" and "export" variables (cf. Sect. 2.1). It is also possible to revise our tool for implementation in other models, as the underlying design concepts are generalizable (cf. Sect. 4.4).

The remainder of the paper is organized as follows: Section 2 introduces EAMv1's code and data structures as well as the features of the model's time integration and output capability that our tool makes use of. Section 3 introduces the key concepts and basic design of our tool. Section 4 describes the implementation of our tool in EAMv1 and Sect. 5 provides a brief user's guide. Section 6 presents three concrete examples to further demonstrate the usage of the tool: a global budget analysis of dust aerosol mass concentration, a composite analysis of sea salt emissions and their dependency on surface wind speed, and a

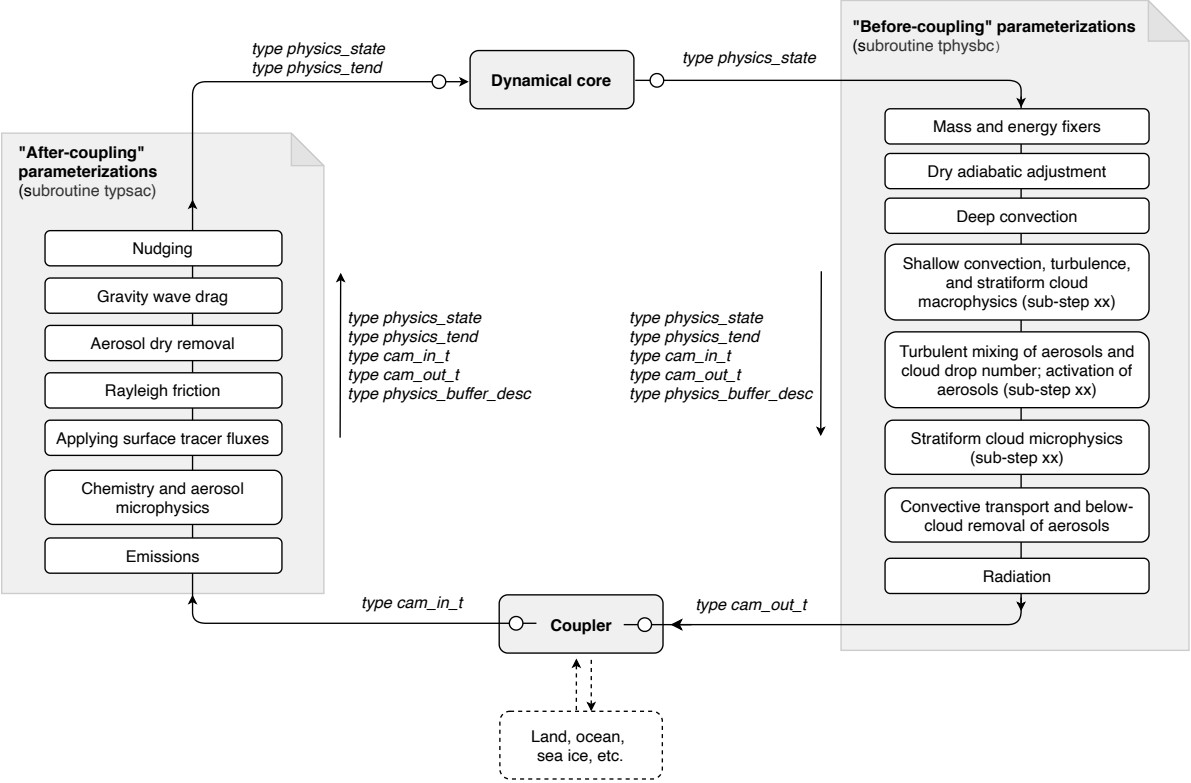

**Figure 1.** A schematic showing the four compartments (gray boxes) of the EAMv1 code: the dynamical core, the coupler, and two groups of parameterizations calculated before or after the communication with the coupler, respectively. The derived data types used for passing information among these compartments and within the two groups of parameterizations are shown in italics. The two small circles shown next to the dynamical core and the two circles placed inside the coupler box represent transfer of information to or from data structures internal to the dynamical core or the coupler. The white boxes with solid outlines shown in the "before-coupling" and "after-coupling" parameterization groups are examples (not complete lists) of parameterizations and numerical treatments included in typical EAM simulations.

conditionally sampled relative humidity budget. Section 7 summarizes the paper and points out possible future improvements and extensions of the tool.

## 2  Host model features

Here, "host model" refers to the AGCM in which our new tool is embedded, in this case EAMv1. We provide some background
information about EAMv1's code structure and data structure in Sect. 2.1 to help explain the implementation and portability of our tool in later sections. We summarize EAMv1's choice of method for coupling atmospheric processes in Sect. 2.2 and

briefly describe how model variables are archived on output files in Sect. 2.3. These features of the host mode are used by our tool.

## 2.1 Data and code structures

EAMv1 is an AGCM consisting of a dynamical core describing the mesh-resolved fluid dynamics and a suite of parameterizations describing various subgrid-scale processes. EAMv1 is also the atmosphere component of the coupled Earth system model E3SMv1 (Golaz et al., 2019) and as such, it communicates with the coupler cpl7 (Craig et al., 2012) to exchange information with the other components of E3SMv1 such as the ocean model, the land model, and the sea ice model, etc. Reflecting both perspectives, the atmosphere model code can be understood as consisting of the four compartments depicted by gray shaded boxes in Fig. 1: the dynamical core, the coupler, and two groups of parameterizations calculated before or after the communication with the coupler, respectively. The driver subroutines for the before-coupling and after-coupling parameterization groups are named `tphysbc` and `tphysac`, respectively (Fig. 1). `tphysbc` and `tphysac` each contains a series of subroutine calls corresponding to various parameterizations. The white boxes with solid outlines in Fig. 1 are examples of such parameterizations. `tphysbc` and `tphysac` also contain code blocks for numerical treatments (e.g., total energy fixers or mass fixers) or for diagnosing quantities of interests. In the remainder of the paper, we refer to those subroutines and code blocks as code compartments, too, although these are sub-compartments of `tphysbc` and `tphysac`. The driver subroutines `tphysbc` and `tphysac` and the code compartments therein are arguably the code units in EAM that researchers of atmospheric physics most often work with.

EAM's dynamical core and physics driver subroutines use different data structures. The following derived data types are defined to pass information among the four compartments shown in Fig. 1 and between parameterizations: The `physics_state` type contains variables describing the atmospheric state that are passed between the physics and dynamics and between parameterizations. Examples of such state variables include air temperature, zonal and meridional winds, vertical velocity, air pressure, pressure layer thickness, geopotential height, surface pressure, and surface geopotential. The `physics_tend` type contains the total tendencies of temperature and horizontal winds caused by all sub-grid processes as well as a few water and energy fluxes that are passed from the parameterization suite to the dynamical core. The import state type `cam_in_t` and export state type `cam_out_t` contain the fields of physical quantities that are provided to EAM by the coupler and to the coupler by EAM, respectively (cf. Chapter 4 in Eaton, 2015). The `physics_buffer_desc` type is defined for constructing the physics buffer that contains fields passed between parameterizations. Dummy variables of these five derived types are available in subroutines `tphysbc` and `tphysac`; collectively, they describe the characteristics of the model atmosphere that vary in space and evolve with time.

A subroutine called by `tphysbc` or `tphysac` may again be a driver for a set of closely related parameterizations and hence calls a number of sub-subroutines. For example, the chemistry driver in `tphysac` has multiple levels of subroutines that correspond to various processes related to chemical gases and aerosol microphysics. Depending on how those lower-level subroutines are organized, EAM variables of the derived type `physics_state` or `physics_buffer_desc` etc. may be available in those lower-level subroutines.

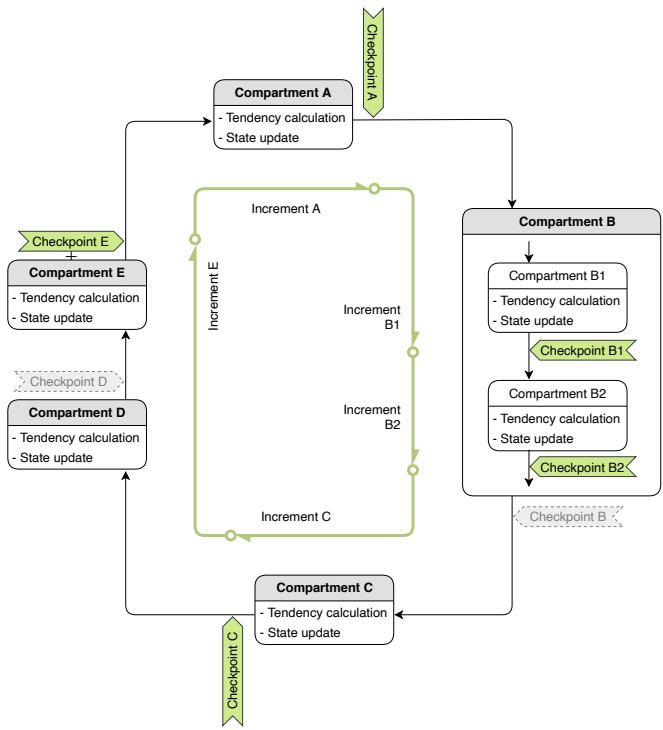

**Figure 2.** A schematic showing a time step of model simulation involving five hypothetical code compartments, A to E, either resolved or unresolved by the model's computational mesh, that are numerically coupled using isolated sequential splitting (cf. Sect. 2.2). Also shown are various tags of locations (referred to as checkpoints, cf. Sect. 3.1) within a time step that are introduced to facilitate diagnostics using the new tool. When the tool is used in a simulation, some checkpoints are activated (i.e., selected by the user and indicated in green here), and others are inactive and indicated in gray. No information is monitored at inactive checkpoints. The green lines with a circle on one end and an arrowhead on the other end depict how increments of model variables are defined. Further details can be found in Sects. 2.2 and 3.1.

## 2.2 Sequential process coupling

EAMv1 solves a set of integral-differential equations to simulate the spatial variation and temporal evolution of the state of the atmosphere. Distinct physical and chemical processes (phenomena) are represented by different compartments of the model code. The primary method used in EAMv1 for coupling those compartments is a method we refer to as isolated sequential
splitting (Fig. 2). In this method, a code compartment produces an estimate of the rate of change of the atmospheric state by considering a single or a set of closely related physical or chemical processes in isolation (i.e., ignoring all other processes represented by other model components). The estimated tendency is used to update the atmospheric state, and then the updated state is passed to the next code compartment. Since EAMv1 has many code compartments, the atmospheric state is updated multiple times within one full time step. Here a full time step is defined as the smallest time integration cycle in which the
effects of all physical processes considered in a simulation have been used to update the model state at least once in advancing

the solution in time. This full time step is often loosely referred to as the "physics time step" in EAMv1 and its predecessors. In a discussion on time stepping and sub-cycling in EAMv1, Wan et al. (2021) referred to the full time steps as the "main process-coupling time steps" and denoted their length by $\Delta t_{\mathrm{CPLmain}}$. The same notation is used in this paper for consistency and clarity. The so-called low-resolution configuration of EAMv1 (with 1 degree horizontal grid spacing) uses $\Delta t_{\mathrm{CPLmain}} = 30$ min by default. Fig. 2 provides a schematic showing a full time step consisting of five hypothetical code compartments labeled as A to E.

A code compartment in EAMv1 might contain sub-compartments that are also connected using the isolated sequential splitting method, like the compartment B depicted in Fig. 2. A concrete example from EAMv1 is deep convection, which consists of the parameterization by Zhang and McFarlane (1995) that describes impact of convective activities on temperature and humidity and a parameterization of the convective momentum transport from Richter and Rasch (2008). These convection-related atmospheric processes are sequentially split within the deep convection parameterization.

Another situation that can also be depicted by the hypothetical compartment B in Fig. 2 is sub-cycling. For example, in EAMv1, the parameterizations of turbulence, shallow convection, and stratiform cloud macrophysics and microphysics are sub-cycled six times within each 30 min full time step. In this case, each sub-cycle can be viewed as a sub-compartment depicted in Fig. 2 (i.e., sub-cycle 1 corresponds to compartment B1, sub-cycle 2 corresponds to compartment B2, etc.).

## 2.3 History output

EAMv1 inherited from its predecessors a flexible mechanism for handling model output (see, e.g., Chapter 8 in Craig et al., 2021). The data files that contain the temporal and spatial distribution of model-simulated physical quantities are called history files. The model can write multiple series of history files with different write frequencies; these series are referred to as history tapes in the source code. Different history tapes can contain different output variables (fields). Whether the values written out should be instantaneous, time-averaged, maximum or minimum during the output time window can be specified for each tape on a field-by-field basis.

The software infrastructure for history output uses internal data types and functions that handle the storage of fields to be written out and perform the calculation of required statistics (e.g., time averages). Typically, researchers focusing on physical or computational aspects of the model do not need to care about the internal workings of this software infrastructure. Rather, they use a subroutine named `outfld` to transfer the values of a model variable to the infrastructure. To provide a context for some descriptions in later sections, we note that while a model variable can change its value multiple times in a time step of $\Delta t_{\mathrm{CPLmain}}$, the value being recorded for output is the snapshot made when the `outfld` subroutine is called. The location in the time integration cycle at which the `outfld` subroutine is called can differ from model variable to variable.

## 3 Nomenclature and design concepts for CondiDiag

We now introduce the key concepts and design features of the new tool. The description in this section is kept general, only referring to EAM when necessary, as the methodology can be applied to or revised for other AGCMs. Details of the implementation in EAMv1 are provided in Sect. 4.

### 3.1 Checkpoints, field values, and increments

In order to discuss the implementation of our tool in the context of the sequential process splitting described in Sect. 2.2, we introduce the following nomenclature:

– A *checkpoint* is a location in the time integration cycle where a snapshot of a model variable can be obtained (cf. Fig. 2). At a checkpoint, the value of a model variable can be retrieved from data structures introduced in Sect. 2.1. Additional quantities can be computed from available variables. Those retrieved or computed variables at the checkpoint can be saved in the data structure specific to our tool and be transferred to the output-handling infrastructure of the standard EAM (cf. Sect. 2.3). If sub-cycles with respect to $\Delta t_{\text{CPLmain}}$ are used, then the end of each sub-cycle is considered to be a different checkpoint.

– The value of a model variable at a checkpoint is referred to as a *field value*. For example, the air temperature after compartment A in Fig. 2 is referred to as the field value of temperature at checkpoint A.

– All checkpoints are inactive by default, meaning no information is retrieved, calculated, or archived by our tool. A checkpoint becomes active when the user selects it at run time (through namelist, cf. Sect. 5.2.3). This flexibility allows a user to focus only on the checkpoints relevant to their specific study; it also saves memory and disk space, as inactive checkpoints will not consume memory or produce information in the model's output files.

– The difference between values of the same model variable at two different checkpoints is referred to as an *increment*. Since there can be inactive checkpoints, an increment calculated by our tool is the difference between the field value at the current checkpoint and the field value at the previous active checkpoint. For example, in Fig. 2, increment E is the difference between checkpoints E and C, with the inactive checkpoint D ignored.

### 3.2 Composite analysis

For a composite analysis, our tool expects the user to specify one or more conditional sampling criteria via run time input (e.g., namelist parameters). The handling of multiple conditions is described later in Sect. 3.3. Here we first explain the handling of a single sampling condition.

During each time integration cycle of length $\Delta t_{\text{CPLmain}}$, values of user-selected variables at active checkpoints are obtained and copied to a data structure internal to our tool. Increments and vertical integrals are calculated if requested. The sampling condition is evaluated at each grid cell in the global domain. Depending on whether the condition is met, the copy of the

user-selected variables in our tool's internal data structure, including their increments and integrals if requested by the user, is assigned either the model-computed values or a fill value, resulting in a conditionally sampled copy. This sampled copy, together with information about the sampling condition, is then transferred to the output handling infrastructure. In the next model time step, the sampling condition is re-evaluated and the user-selected model variables re-sampled. The details are explained below.

### 3.2.1 Defining a condition

A key element of a sampling strategy is the atmospheric condition to be used to categorize data. Necessary elements in the definition of a condition include (1) a *metric* (which can be any 2D or 3D field, e.g., air temperature or surface pressure), (2) a *threshold* (which is a number, e.g., -40 °C or 500 hPa), and (3) a *comparison type* (e.g., smaller than or equal to). In our tool, a metric can be any prognostic or diagnostic variables in the host model or a quantity that can be diagnosed from existing variables. Currently supported comparison types include (i) $<$, (ii) $\leqslant$, (iii) $>$, (iv) $\geqslant$, and (v) equal to within a tolerance. Type (v) can be used to select values within a range. For example, choosing a threshold of -20°C and a tolerance of 20 °C would allow the user to sample grid cells with air temperature between -40°C and 0 °C. The user's choices of metric, threshold, comparison type, and tolerance (if applicable) are expected to be specified through run time input.

Another key element of the definition of the sampling condition is the location in the time integration cycle at which the sampling condition should be evaluated. As explained earlier in Sect. 2.2, the atmospheric state defined by the prognostic variables of EAM's governing equations is updated multiple times within one full time step of $\Delta t_{\mathrm{CPLmain}}$ due to the sequential splitting method used for process coupling. For diagnostic quantities (e.g., relative humidity), the values consistent with the prognostic state also evolve within each time step even though the arrays in the programming language can temporarily contain inconsistent values until the next time of calculation. Because of such evolutions within a time step, our tool requires the user to specify at which checkpoint (cf. Sect. 3.1) a sampling condition should be evaluated. The implementation of this aspect in EAMv1 is discussed in more detail in Sect. 5.2.1. Also because of the evolution of model variables within a time step, one needs to be cautious when obtaining values of diagnostic quantities for use by our diagnostic tool. This point is further explained in Sect. 4.1.2.

### 3.2.2 Condition metric and field of flags

In this first version of our tool, the metric used in defining a sampling condition can be one of the following types of model variables:

- a 2D field that covers the entire horizontal domain of the model, such as the surface pressure or total cloud cover;

- a 3D field defined at layer mid-points or as layer averages, e.g., air temperature, cloud fraction, or the mass mixing ratio of a tracer in EAMv1,

- a 3D field defined at the interfaces between adjacent layers, e.g, the convective mass flux predicted by the deep convection parameterization or the net longwave radiative flux calculated by the radiation scheme in EAMv1.

For each condition metric, a flag field with the same spatial dimensions is defined in the data structure internal to our tool. After a sampling condition is evaluated at a grid cell in the 2D or 3D domain, the flag field is assigned a value of 1 if the condition is met and a value of 0 otherwise. The flag field, when averaged over time, equals the frequency of occurrence of meeting the sampling condition at each individual grid cell. The flags at different grid cells can be averaged in space, either over the entire 2D or 3D domain or over a subdomain, to calculate the frequency of occurrence of the sampling condition in the corresponding domain, but the spatial averages are expected to be done during post-processing instead of during model integration. A use case example involving both temporal and spatial averaging can be found in Sect. 6.3.

After the sampling condition is evaluated over the entire 2D or 3D domain, the condition metric itself is sampled, meaning that the field of values transferred to the output-handling software contains the model-computed values where the condition is met and a fill value of zero where the condition is not met. In other words, the masking indicated by the flag field is applied to the condition metric as well. Recall that the output-handling infrastructure of EAM supports both instantaneous and time-averaged model output. Since EAM is a climate model, time-averaged output is expected to be used more often. Our tool uses a fill value of zero for archiving the condition metric and the other monitored model variables to make sure that time steps in which the sampling condition is not met make zero contributions to the time average. Later on, during post-processing, when a time average of a condition metric is divided by the time average of the corresponding flag, we get the composite mean, i.e., the average over the time steps when the condition is met.

### 3.2.3   Monitored model variables

Our tool allows for multiple model variables to be monitored under the same sampling condition. To distinguish those monitored variables from the condition metric, the monitored variables are referred to as the quantities of interest (QoIs) in the remainder of this paper and in our code. QoIs monitored under the same condition can have different vertical dimension sizes:

- When the QoI has the same dimension size as the condition metric, the masking indicated by the flag field can be applied in a straightforward manner.

- If the metric is 2D and the QoI is 3D, then the same 2D masking is applied to all vertical layers or interfaces.

- If the metric is 3D and the QoI is 2D, then a grid cell in the 2D domain is selected if any layer midpoint or interface in that column is selected. For example, to quantify the shortwave cloud radiative effect (the QoI) in the presence of ice clouds, one can choose a sampling condition of non-zero ice crystal concentration. Then, if ice crystals occur in any layer in a grid column, then the shortwave cloud radiative effect of that grid column will be sampled.

Like the archiving of the condition metric explained in Sect. 3.2.2, a QoI gets a fill value of zero at grid cells where the condition is not met, so that the composite mean can be derived by dividing the time-averaged QoI by the time-averaged flag field.

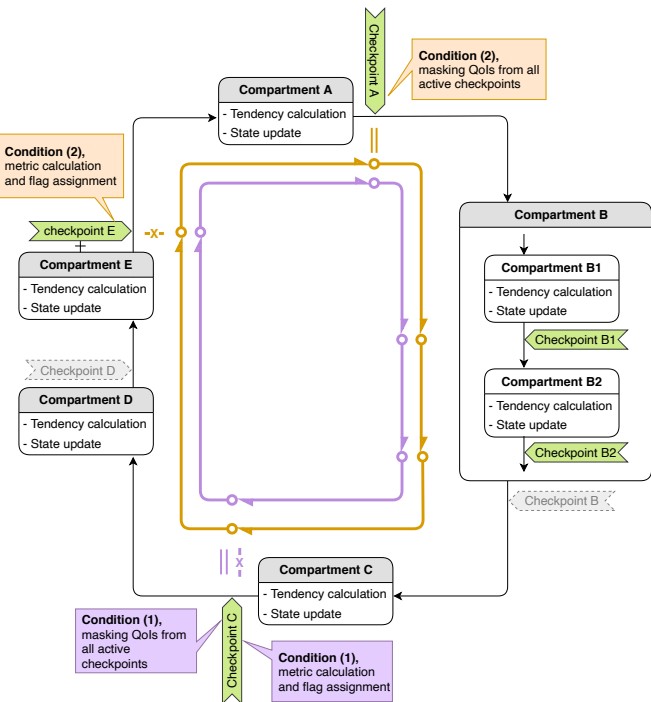

**Figure 3.** A schematic showing two sampling conditions indicated in brown and purple. The "-X-" marks indicate locations in the time integration cycle where the condition metrics are evaluated. The double-bars indicate the end of validity of the evaluated sampling conditions. More details can be found in Sect. 3.2.4. Like in Fig. 2, green tags are active checkpoints being monitored by the tool. Gray tags with dashed borderlines are inactive checkpoints, which are ignored in the simulation.

### 3.2.4 Time window of validity of an evaluated condition

Our tool is designed to evaluate a sampling condition once per each $\Delta t_{\text{CPLmain}}$ at a user-specified checkpoint X and the tool can monitor QoIs at multiple checkpoints within $\Delta t_{\text{CPLmain}}$. By default, the masking resulting from a condition evaluated at checkpoint X is applied retrospectively to all active checkpoints from X until just before the previous encounter of X (i.e., X in the previous time step). This is illustrated by condition (1) shown in purple in Fig. 3, where the sampling condition is evaluated at checkpoint C and the masking is applied retrospectively to checkpoints B2, B1, A, and E.

To provide more flexibility, our tool also allows the user to specify a different checkpoint as the end-of-validity mark for a sampling condition, which we indicate with double-bars in Fig. 3. A hypothetical example is given as condition (2) shown in brown in the figure. There, the end-of-validity mark (brown double-bar) is placed at checkpoint A while the sampling condition is evaluated at checkpoint E. The masking determined at E is applied to E and the subsequent checkpoint A, as well as retrospectively to checkpoints C, B2, and B1 before E. An example from EAMv1 showing such a usage can be found in Sect. 6.3.

### 3.3 Multiple sampling conditions in one simulation

A single sampling condition is defined by a combination of (i) a metric, (ii) a threshold, (iii) a comparison type, (iv) a tolerance if the comparison type is "equal to", (v) a condition-evaluation checkpoint, and (vi) an end-of-condition-validity checkpoint. Changing any of these elements will result in a new sampling condition. Our tool allows for multiple conditions to be used in a single simulation (cf. Fig. 3), and these multiple sampling conditions can use different condition metrics.

For software simplicity, the current implementation only allows one and the same set of QoIs and checkpoints to be monitored under all sampling conditions. In the example illustrated in Fig. 3 where two conditions, (1) and (2), and five active checkpoints (A, B1, B2, C, and E) are activated, let us assume the user has chosen to monitor four QoIs, $T$, $q_v$, $u$, and $v$. The same four QoIs and five checkpoints will be monitored for both sampling conditions. The current implementation does not allow, for example, monitoring only $T$ and $q_v$ at checkpoint A and C under condition (1) and only $u$ and $v$ at checkpoints A, B1, and B2 under condition (2), although this kind of flexibility can be considered for future versions of CondiDiag if needed.

Since the current implementation monitors the same QoIs for all sampling conditions in the same simulation, one can run into a situation where the metric and the QoI are both 3D fields but have different numbers of vertical layers (e.g., the metric is the air temperature defined at layer midpoints while the QoI is the net longwave radiative flux defined at layer interfaces). In such a case, masking will be skipped, meaning this specific QoI will be captured for output as if no conditional sampling had happened.

### 3.4 Mass-weighted vertical integral of QoIs

For spatially 3D QoIs defined at layer midpoints or as cell averages, the vertical integral weighted by air mass can be calculated during the time integration and then conditionally sampled and written out as 2D variables. This applies to both field values and their increments.

One note of caveat is that in EAM's physics parameterizations, the mixing ratios of water species (vapor, cloud liquid and ice, rain and snow) are defined relative to the mass of moist air (i.e., dry air plus water vapor) while the mixing ratios of aerosols and chemical gases are defined with respect to dry air. Our tool expects the user to specify which kind of air mass (moist or dry) should be used for each QoI when vertical integrals is requested (cf. Sect. 5.2.4).

### 4 Implementation in EAMv1

This section explains how the design features described in Sect. 3 are implemented in EAMv1. We provide an overview of the new Fortran modules added specifically for the tool (Sect. 4.1), introduce a general-purpose diagnostics module (Sect. 4.2), and summarize the changes made to the original EAMv1 code (Sect. 4.3). We keep these sections brief but provide two versions of the EAMv1 code on Zenodo (DOI: 10.5281/zenodo.5530188) corresponding to the GitHub commits before and after the implementation of CondiDiag1.0, so that readers can review the details of the code changes if needed. The Zenodo archive also

contains a third tar ball containing only the source files that were added or revised during the implementation of CondiDiag1.0, as well as a copy of the original version of the revised EAMv1 files.

## 4.1 CondiDiag-specific new modules

Four new modules are added to define data structures and support key functionalities of our diagnostic tool. These are briefly described below.

### 4.1.1 Data structure module

The module `conditional_diag` contains definitions of the basic data structures used by our tool and subroutines for initializing the corresponding Fortran variables.

A Fortran variable `cnd_diag_info` of the derived type `cnd_diag_info_t` contains the metadata that describes the user's conditional sampling strategy and budget analyses configuration. A namelist `conditional_diag_nl` (cf. Sect. 5.2) is also defined in this module, and a subroutine `cnd_diag_readnl` parses the user's namelist input and populates the information to `cnd_diag_info`.

A second derived type `cnd_diag_t` is defined for storing the values of the metrics, flags, and the field values and increments of QoIs. The corresponding Fortran variable is an array named `phys_diag`; the array is defined in a different module (explained in Sect. 4.3.1). The subroutines that allocate memory for elements of `phys_diag` and their components are included in module `conditional_diag`.

### 4.1.2 Key algorithm module

The module `conditional_diag_main` contains the key subroutine of our tool, named `cnd_diag_checkpoint`, which obtains the values of the condition metrics and QoIs, calculates the QoI increments, evaluates the sampling conditions, applies conditional sampling, and transfers the sampled fields to the output-handling infrastructure of EAM. Examples showing how the subroutine is invoked in EAM is explained in Sect. 4.3.2.

As mentioned earlier in Sect. 3.1, the condition metrics and QoIs can be existing components of EAM's state variable, physics buffer, and the atmosphere import and export data structures (cf. Sect. 2.1; note that the `physics_tend` type is not used by our tool). For example, air temperature is a component of the atmosphere state variable; hence the values are retrieved in subroutine `get_values` in module `conditional_diag_main` by

```
case('T')
  arrayout(1:ncol,:) = state%t(1:ncol,:)
```

Condition metrics and QoIs can also be physical quantities that need to be calculated from components of EAM's existing data structures. For example, the relative humidity with respect to ice is obtained by

```
case ('RHI')
```

```
      call relhum_ice_percent(  &
          ncol, pver,          &! intent(in)
state%t(:ncol,:),    &! intent(in)
          state%pmid(:ncol,:), &! intent(in)
          state%q(:ncol,:,1),  &! intent(in)
          arrayout(:ncol,:)    )! intent(out)
```

In these examples, "T" and "RHI" need to be unique names within the module `conditional_diag_main`; these will
also be the metric or QoI names that the users refer to in the namelist `conditional_diag_nl` (cf. Sect. 5.2). The currently
implemented metric and QoI names are listed in Table A1 in Appendix A. Additional metrics and QoIs can be added following
the existing examples. We note that some of the variable names in Table A1 coincide with EAM's standard history variable
names but the coincidence has no significance. Because a QoI can be monitored at different checkpoints and under different
conditions, those different combinations will each correspond to a distinct variable name in the history files, as explained in
Sect. 4.1.3.

Here, it is worth pointing out one important caveat for obtaining values of diagnostic quantities in the host model. As
mentioned in Sect. 3.2.1, the values of diagnostic quantities that are consistent with the prognostic state effectively evolves
within a full model time step but the arrays in the programing language might have only one or a few updates per full time
step and hence can temporarily have inconsistent values. Care is needed to handle the corresponding code blocks in subroutine
`get_values` of module `conditional_diag_main`. Let us assume the host model has a diagnostic quantity whose value
is saved in the physics buffer under the name `ABC`.

If the user's intention is to *understand the host model's code* by tracking *when* the physics buffer's component `ABC` is updated
within a full model time step, a code block like the following is needed:

```
      case('ABC'//'_PBUF')
idx = pbuf_get_index('ABC')
        call pbuf_get_field( pbuf, idx, ptr2d)
        arrayout(:,:) = ptr2d
```

If the user's intention is to *understand the physics* by monitoring the values of `ABC` that are consistent with the evolving
prognostic state, a code block like the following is needed which recalculates the value of `ABC` from the state variable:

```
case('ABC'//'_EVOL')
        call calculate_abc( state, ..., arrayout)
```

The RHI budget example shown in Section 6.3 falls into the second category.

### 4.1.3   History output module

The module `conditional_diag_output_utils` is responsible for adding the following items to EAM's master list of
history output variables:

- the conditionally sampled metric field named with the pattern `cnd<index>_<metric_name>` where `<index>` is a two-digit number (e.g., `cnd01_T` if the first sampling condition uses air temperature as the metric);

- the flag field (see Sect. 3.2.2) named `cnd<index>_<metric_name>_flag`;

- one output variable corresponding to each QoI at each active checkpoint under each sampling condition, named with the pattern `cnd<index>_<QOI_name>_<checkpoint_name>`. For example, `cnd01_CLDLIQ_DYNEND` is the stratiform cloud liquid mixing ratio monitored at checkpoint DYNEND under condition 1. If increments of the QoI are calculated and archived, these will be named similar to the QoIs but with a suffix `_inc` append, e.g., `cnd01_CLDLIQ_DYNEND_inc` for the increment of CLDLIQ at checkpoint DYNEND under condition 1.

- If the mass-weighted vertical integral is requested for a QoI, then a suffix `_v` will be appended to the QoI name. For example, `cnd01_CLDLIQ_v_DYNEND` is the column burden of CLDLIQ at checkpoint DYNEND under condition 1 and `cnd01_CLDLIQ_v_DYNEND_inc` is the corresponding increment.

We expect that users of our tool should not need to touch the `conditional_diag_output_utils` module unless they want to revise the naming conventions for variables in the history files.

It is worth noting that for any of the output variables added by our tool, EAM's standard history output functionalities apply (cf. Sect. 2.3). For example, each variable can be added to or excluded from one or multiple history tapes and be written out at the user-specified frequencies. For temporal statistics, both instantaneous and time-averages can be used in the current implementation. Maximum and minimum values etc. need to be used with care as unselected grid cells are filled with zeros. In future versions, we will consider allowing the user to specify what missing value should be assigned to each QoI.

### 4.1.4 Restart module

Because our diagnostic tool uses its own data structure, new subroutines have been included to add additional contents to EAM's restart files. These subroutines are placed in the module `conditional_diag_restart`. As long as a user does not change the data structures defined in module `conditional_diag`, there should be no need to touch the restart module even if they add new metrics and QoIs to the key algorithm modules `conditional_diag_main` and `misc_diagostics`.

### 4.2 General-purpose diagnostics module

We imagine a user might want to provide their own subroutines to calculate new metrics or QoIs that are not available in the host model or recalculate diagnostic quantities to obtain values consistent with the evolving prognostic state (like `relhum_ice_percent` in the code snippet in Section 4.1.2). In such cases, we recommend those subroutines be placed in the module `misc_diagnostics` rather than in `conditional_diag_main`, because we view those user-provided subroutines as general-purpose diagnostic utilities that could also be used by other parts of EAM (e.g., in some parameterizations for diagnostic purposes).

### 4.3 Other code changes in EAMv1

Apart from adding the five modules explained in Secitons 4.1 and 4.2, the implementation of our tool in EAMv1 only involved a very small number of code changes, as described below.

#### 4.3.1 The `phys_diag` array and its elements

Our tool uses its own derived data type `cnd_diag_t` for storing values of the condition metrics, flags, and the field values, increments, and vertical integrals of QoIs (cf. Sect. 4.1.1). The data storage closely follows the handling of EAM's model state variable.

To explain the background, we note that in order to parallelize the parameterization calculations on supercomputers, EAM's global domain is divided into "chunks" of grid columns. A chunk contains a compile-time configurable number of columns
that are not necessarily neighbors in the geographical sense. Each call of `tphysbc` or `tphysac` performs calculations of the corresponding parameterizations in a single grid chunk, while each MPI (Message Passing Interface) process on the super-computer typically performs calculations for multiple chunks (cf. Chapter 4 in Eaton, 2015). In `tphysbc` and `tphysac`, the dummy variable `state` is declared as a scalar of type `physics_state`, and this scalar contains data for a single grid chunk. The parent routines of `tphysbc` and `tphysac` declare rank-one arrays (of type `physics_state`) named `phys_state`
for storing data for all chunks handled by the same MPI process.

Similarly, for implementing our tool in EAMv1, rank-one arrays of type `cnd_diag_t` named `phys_diag` are declared in parent routines of `tphysbc` and `tphysac`. The scalar variable of type `cnd_diag_t` in `tphysbc` or `tphysac` is named `diag`.

#### 4.3.2 Checkpoints

The checkpoints listed in Tables B1 and B2 in the Appendices have been added to `tphysbc` and `tphysac` by inserting code lines like

```
call cnd_diag_checkpoint( diag,      &! inout
     'DYNEND', state, pbuf,     &! in
     cam_in, cam_out            )! in
```

(These code lines are inserted after the white boxes with solid outlines shown in Fig. 1.) Here, `diag` is the scalar variable of type `cnd_diag_t` explained in Sect. 4.3.1; "DYNEND" is the unique string identifying this checkpoint; `state`, `cam_in` and `cam_out` are scalar variables of derived types declared in the original EAM code.

As mentioned in Sect. 2.1, the EAM code has a hierarchical structure; the `state`, `cam_in` and `cam_out` variables as well as the physics buff are available also in some lower-level routines called by the physics drivers `tphysbc` and `tphysac`.
Checkpoints can be added to those lower-level routines. To demonstrate this point, in CondiDiag1.0, checkpoints have been included in the stratiform cloud macrophysics driver subroutine `clubb_tend_cam` in the form of, e.g.,

```
call cnd_diag_checkpoint( diag,      &! inout
     'CLUBB'//char_macmic_it,       &! in
     state1, pbuf, cam_in, cam_out )! in
```

where the character string `char_macmic_it` labels the sub-steps within a full time step $\Delta t_{\mathrm{CPLmain}}$. It is worth emphasizing that `state1` (instead of `state`) is referred to in the code snippet quoted above because `state1` is the atmospheric state variable that is sequentially updated by various code blocks (compartments) in `clubb_tend_cam`.

## 4.4   Portability

Our new tool was originally developed for and implemented in version 1 of EAM and was then tested in v2 and some in-
between versions. The porting turned out to be straightforward as the basic code and data structures in EAM had not changed. To implement CondiDiag in models outside the EAM/CAM model families will require some significant adaptation. Some thoughts are shared here.

We assume the host model has a few high-level driver subroutines (or one driver) that organizes code compartments corresponding to various atmospheric processes. This, to our knowledge, is common in AGCMs.

Our code also makes use of the fact that the drivers use derived data types to organize a large number of model variables of interest for physics-oriented or numerics-focused studies. These derived data types make our code more flexible and compact, especially for conditional sampling.

For performing budget analysis, our current algorithm assumes the sequential splitting method is used in the host model. For models that use different coupling methods (e.g, parallel splitting or a mixture of methods), it might be possible to obtain the
budget terms directly from the tendencies saved in existing model variables.

The four new modules CondiDiag introduces to EAM (cf. Sect. 4.1) all use some EAM-specific data structures and software functionalities. For porting to a new model, some parts of these modules will be straightforward to port and the other parts will need a rewrite.

The `conditional_diag` module has the weakest dependency on EAM. The meta-data handling part (i.e., parsing the
user's choices of QoIs, metrics, etc.) is independent of EAM's data structures. The module also contains a few subroutines that allocate memory for the derived-type arrays used for storing the QoIs, metrics, etc.. The code therein assumes a chunk-based domain decomposition, which likely will need to be adapted to the new host's data structure.

The `conditional_diag_main` module contains subroutines for retrieving field values, deriving increments, calculating vertical integrals, and performing conditional sampling, etc. The subroutines assume all QoIs and condition metrics can be
retrieved or recalculated from EAM-specific data structures described in Sect. 2.1, hence the dummy variables and their usage will need to be adapted for a new host model.

Module `conditional_diag_output_utils` and module `conditional_diag_restart` will each need a rewrite for a new host. The key task of the subroutines therein is to do I/O for all components of the derived type `cnd_diag_t`. We expect that one needs to follow the host model's way of handling I/O for 2D and 3D variables. The rewrite will likely be
somewhat tedious but presumably not difficult.

## 5 User's guide

The new tool is expected to be useful for a wide range of simulations routinely performed by the model developers and users, including debugging simulations that are a few time steps long, short few-day simulations for preliminary testing or weather forecast style simulations for comprehensive evaluations of the model physics following protocols like Transpose-AMIP (e.g., Phillips et al., 2004; Williams et al., 2013; Williamson et al., 2005; Martin et al., 2010; Xie et al., 2012; Ma et al., 2013, 2014), as well as more traditional multi-year to multi-decade simulations.

To obtain process-level understanding of model behavior, it can be useful to use the new tool in an iterative manner. For example, for a study like Zhang et al. (2018) where one needs to identify code compartments that result in negative values of specific humidity, we can start by carrying out a few-day or one-month simulation with unconditional sampling, choosing a large number of checkpoints to monitor code compartments that are expected to affect humidity or might inadvertently do so because of computational artifacts or code bugs. We let the tool diagnose and archive time averages of the specific humidity and its increment at these checkpoints to get a sense of typical values of the state variable and identify sources and sinks of moisture. In a second step of investigation, we eliminate from the previous selection any checkpoints that have been confirmed to not show humidity change in any grid cell or time step in the few-day or one-month simulation. From the shorter list, we can pick one or multiple code compartments as suspected culprits of negative specific humidity. If $m$ suspects are selected for further investigation, then $m$ sampling conditions can be specified in the next simulation, all using $q_v < 0$ as the sampling criterion but each evaluated after a different suspect. We can also select some QoIs (e.g., temperature, specific and relative humidity, wind, total cloud fraction, cloud liquid and ice mixing ratios, etc.) to monitor both right before and right after the code compartments that are suspected to cause negative water vapor. We can request both the field values and increments of these QoIs to be archived, as time averages or instantaneous values, or both. This second step might provide useful clues of the typical meteorological conditions under which negative water vapor is predicted in the model. If pathological conditions are identified, then we can carry out additional simulations using relevant sampling conditions to further investigate the causes of those pathologies.

This section explains how investigations described above can be performed using our tool. We first present a typical workflow in Sect. 5.1 to illustrate the steps that a user needs to go through when designing an analysis and setting up an EAM simulation using our tool. We then explain the namelist parameters of our tool in Sect. 5.2.

### 5.1 User workflow

The schematic in Fig. 4 summarizes the steps to take for designing a composite or budget analysis using our tool. It also points to relevant concepts explained in earlier sections and namelist parameters explained below.

### 5.2 Namelist `conditional_diag_nl`

Users specify their conditional sampling and budget analysis strategy via the namelist `conditional_diag_nl`, which consists of five groups of parameters.

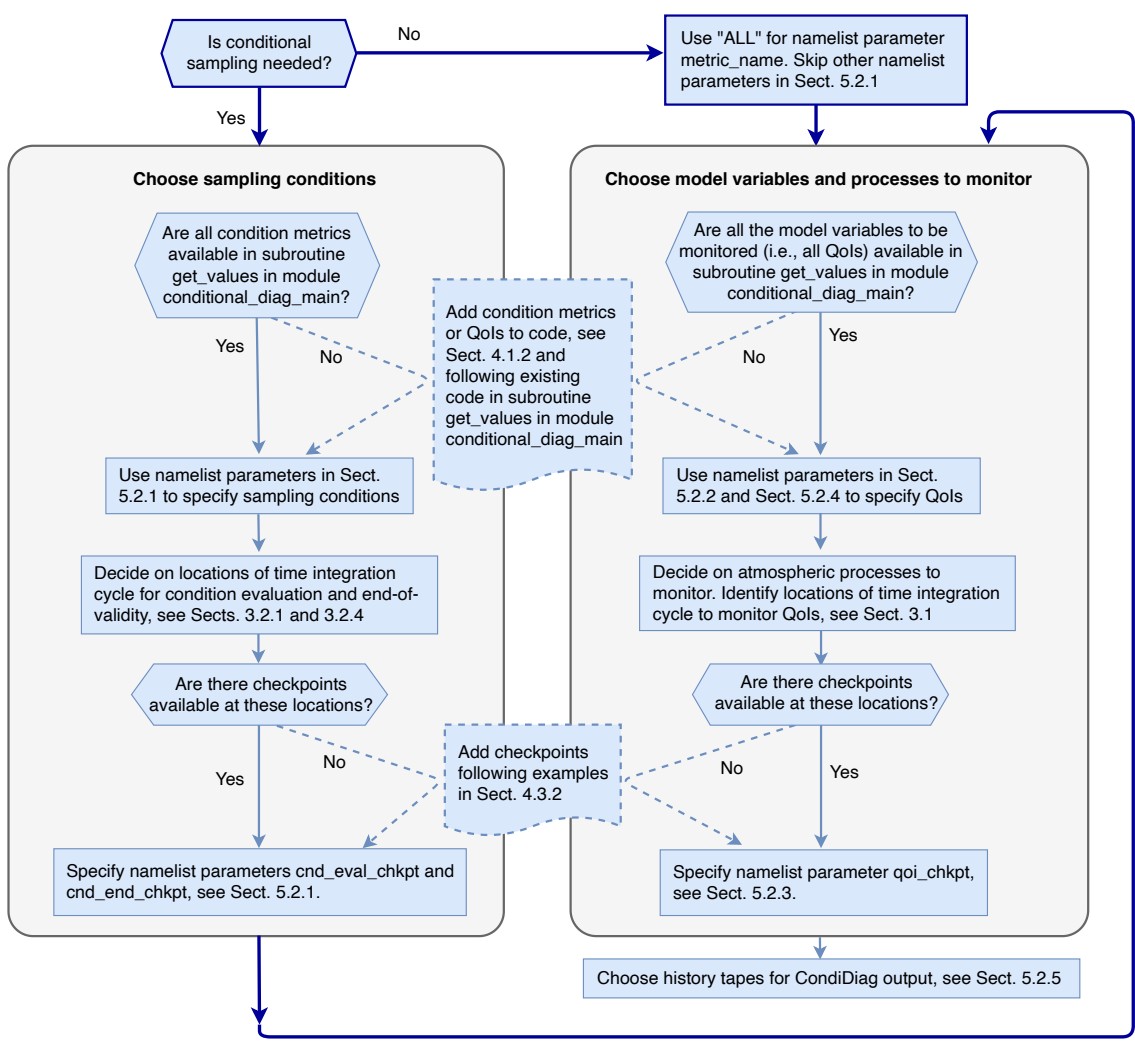

**Figure 4.** A schematic showing the typical steps a user needs to go through for setting up an EAM simulation with online conditional sampling or budget analysis enabled using our tool. Dashed lines indicate places where code changes or additions are needed from the user.

### 5.2.1 Specifying sampling conditions

For the specification of sampling conditions, we have

– metric_name, a character array containing the names of the condition metrics to be used in a simulation;

    – metric_nver, an integer array specifying the number of vertical levels of each metric. This is meant to help distinguish physical quantities that (1) have no vertical dimension, (2) are defined at layer mid-points, and (3) are defined at layer

interfaces. Valid values for metric_nver are 1, `pver` (e.g., 72), and `pverp` (e.g., 73), where `pver` and `pverp` are EAM's variable names for the number of vertical layers and interfaces, respectively.

– `metric_cmpr_type`, an integer array specifying the types of comparison to be used for each condition (one entry per condition): 0 for "equal to within a tolerance", 1 for "greater than", 2 for "greater than or equal to", -1 for "less than", and -2 for "less than or equal to";

   – `metric_thereshold`, a double-precision floating-point array specifying the threshold values that the metrics will be compared to (one threshold for each condition);

– `metric_tolerance`, a double-precision floating-point array specifying the tolerances for conditions with comparison type 0 (one tolerance for each condition; the value will have no effect for conditions with a non-zero comparison type);

   – `cnd_eval_chkpt`, a character array specifying at which checkpoints the conditions will be evaluated (see Sect. 3.2.1; one checkpoint for each condition).

   – `cnd_end_chkpt`, a character array specifying the checkpoints defining the end of validity of an evaluated condition
(see Sect. 3.2.4; one checkpoint per condition). If not specified by user, the end-of-time-step checkpoint will be set to the condition-evaluation checkpoint (`cnd_eval_chkpt`).

### 5.2.2   Specifying monitored model variables

The QoIs to be monitored are specified via a character array `qoi_name`. The number of vertical levels of each QoI is specified through the integer array `qoi_nver`. If no QoIs are specified but some sampling condition have been chosen, then conditional
sampling will only be applied to the metrics.

   The monitoring of QoI field values are turned on by the logical scalar `l_output_state`. A second logical scalar, `l_output_incrm`, is used to turn on or off the monitoring of QoI increments. Users' choice for the two switches will be applied to all QoIs.

### 5.2.3   Choosing checkpoints

The checkpoints at which the QoIs will be monitored are specified by a character array `qoi_chkpt`. The sequence in which they are mentioned in the namelist has no significance. Note that the same checkpoints are applied to all QoIs. Also note that if the user specifies a checkpoint name that does not match any checkpoint implemented in the code (e.g., because of a typographical error), then our tool will act as if the wrong checkpoint is an inactive one - in the sense that it will get ignored when the tool attempts to obtain QoI field values and calculate increments as the simulation proceeds; the history files will
contain output variables corresponding to the incorrect checkpoint name but those output variables will contain zeros.

### 5.2.4 Turning on vertical integral

The calculation of mass-weighted vertical integrals of QoIs can be enabled by the integer array `qoi_x_dp`. The values are specified in relation to `qoi_name`, i.e., one value of `qoi_x_dp` for each QoI. 0 is interpreted as no integral; the QoI will be sampled and written out as a 3D field. If 1 (moist) or 2 (dry) are selected, the corresponding (moist or dry) air mass will be used for vertical integral of that QoI at all active checkpoints in the simulation.

If the user wishes to monitor both a 3D QoI and its vertical integral, they can specify the same QoI twice in `qoi_name` and set one of the corresponding elements in the `qoi_x_dp` array to 0 and the other to an appropriate value (1 for moist and 2 for dry) to request vertical integral. A use case example is provided in Sect. 6.1.

For budget analyses in which mass conservation error is an important topic, there are some nuances related to the fact that the aerosol and chemical gas mixing ratios are converted from drying mixing ratios to moist mixing ratios close to the end of `tphysac`. This is further discussed in Appendix C.

### 5.2.5 Turning on history output

A user might want to write out multiple copies of the conditional diagnostics or budget diagnostics to different history files corresponding to different output frequencies or temporal averaging. To support such needs, the integer array `hist_tape_with_all_output` specifies which history files will contain the full set of output variables from our tool. For example, `hist_tape_with_all_output = 1, 3` will include the output to the h0 and h2 files. Again, we note that the standard output functionalities in EAM explained in Sect. 2.3 still apply.

### 5.3 Using unconditional sampling

One of the main motivations for creating our tool is to facilitate budget analysis. If an analysis is to be carried out for the entire computational domain and all time steps, then a special metric named `ALL` can be used. In such a case, the user can ignore (skip) the other namelist parameters in 5.2.1. When `ALL` is used, the condition evaluation will be skipped during the model's integration (see example in Sect. 6.1). Another way to use unconditional sampling is to specify a condition that will always be fulfilled (e.g., relative humidity higher than -1%). A use case example is provided in Sect. 6.3).

## 6 Use case examples

This section demonstrates the usage of the new tool using three concrete examples:

The first example is a global budget analysis without conditional sampling. It demonstrates how to request unconditional sampling and how to request that increments of model variables be calculated and archived as time averages. This first example also demonstrates that with our tool, it is convenient to obtain both vertical profiles and vertical integrals of the budget terms.

The second example is a composite analysis without budget terms. It demonstrates how to use multiple sampling conditions in the same simulation and also shows that the tool can be used to perform a univariate probability distribution analysis.

In the third example, the increment diagnosis and conditional sampling capabilities are combined to perform a conditional budget analysis. The example demonstrates how metrics and monitored QoIs can be chosen to be physical quantities that need to be calculated from the host model's state variables using user-provided subroutines.

The examples shown here use 1-month simulations of October 2009 with monthly (or monthly and daily) output. All simulations were carried out with active atmosphere and land surface as well as prescribed sea surface temperature and sea ice concentration, at 1 degree horizontal resolution with out-of-the-box parameters and time integration configurations of EAMv1.

## 6.1 A global budget analysis of dust aerosol mass mixing ratio and burden

The first example is a global dust aerosol mass budget analysis without conditional sampling. The simulation is designed to provide insight into the atmospheric processes that change the burden (vertical integrals) of dust aerosols in two size ranges (accumulation mode and coarse mode). In particular, we are interested in dust emission, dry removal (i.e., sedimentation and dry deposition at the Earth's surface), resolved-scale transport, subgrid-scale turbulent transport and activation (i.e., nucleation scavenging), as well as the wet removal caused by precipitation collecting particles by impaction, and resuspension caused by evaporation of precipitation.

**Table 1.** Namelist setup used in the dust budget analysis example in Sect. 6.1.

```
metric_name = 'ALL'

qoi_chkpt = 'CFLXAPP', 'AERDRYRM',
            'PBCINI',  'STCLD',    'AERWETRM'

qoi_name = 'dst_a1', 'dst_a1', 'dst_a3', 'dst_a3'
qoi_nver =  72,       72,       72,       72
qoi_x_dp =  0,        2,        0,        2

l_output_state = .false.
l_output_incrm = .true.

hist_tape_with_all_output = 1,  2
nhtfrq                     = 0, -24
mfilt                      = 1,  31
```

### 6.1.1 Simulation setup

The namelist setup for this study is shown in Table 1. Only one condition is specified: the special metric `ALL` is used to select the entire model domain and all time steps.

The QoI names `dst_a1` and `dst_a3` are EAM's tracer names for dust mass mixing ratio in the accumulation mode and coarse mode, respectively. Each tracer name is mentioned twice under `qoi_name`, with corresponding `qoi_x_dp` values of 0

**Table 2.** For the dust budget analysis example in Sect. 6.1: atmospheric processes corresponding to increments diagnosed at the checkpoints selected in the namelist shown in Table 1.

| Checkpoint | Atmospheric processes |
|---|---|
| CFLXAPP | Surface fluxes of aerosol and chemical tracers |
| AERDRYRM | Dry removal of aerosols |
| PBCINI | Resolved transport |
| STCLD | Turbulent mixing and aerosol activation |
| AERWETRM | Wet removal and resuspension of aerosols |

and 2, meaning that both the vertical distribution of the tracer and its column burden are monitored. With `l_output_state`
set to `.false.` and `l_output_incrm` set to `.true.`, the tool captures the dust mass mixing ratio increments caused by the targetted atmospheric processes but not the mixing ratios. Five checkpoints are chosen for monitoring the dust budget. The corresponding atmospheric processes are listed in Table 2. (We remind the users that, as shown in Fig. 2, the code compartments that contribute to increments diagnosed at a checkpoint not only depends on where this checkpoint is located in the time integration cycle but also where the previous active checkpoint is located.)

The full set of fields tracked by our tool are sent to output files 1 (the h0 file) and 2 (the h1 file), with the h0 file containing monthly averages and the h1 file containing daily averages.

### 6.1.2  Results

Figure 5 shows a one-month mean geographical distribution of the sources and sinks of dust mass in the coarse mode (unit: kg m$^{-2}$ s$^{-1}$). The values shown are the output variables `cnd01_dst_a3_v_<checkpoint_name>_inc` in the h0 file
divided by $\Delta t_{\text{CPLmain}}$ = 30 min. Figure 6 shows examples of the globally averaged vertical profiles of the coarse-mode dust mass mixing ratio tendencies (unit: kg kg$^{-1}$ s$^{-1}$). The black curves are monthly averages. The colored horizontal bars indicate variability of the daily averages derived from the 3D increment fields `cnd01_dst_a3_<checkpoint_name>_inc` written to the h1 file.

### 6.2  A composite analysis of sea salt emissions in relation to surface wind speed

This example demonstrates the use of composite analysis (without budget terms) to provide insight into wind speed impacts on emission fluxes of sea salt aerosol in various size ranges. The intension is to examine the geographical distribution of sea salt emission fluxes under weak, medium, and strong wind conditions and quantify their relative contributions to the total emission fluxes.

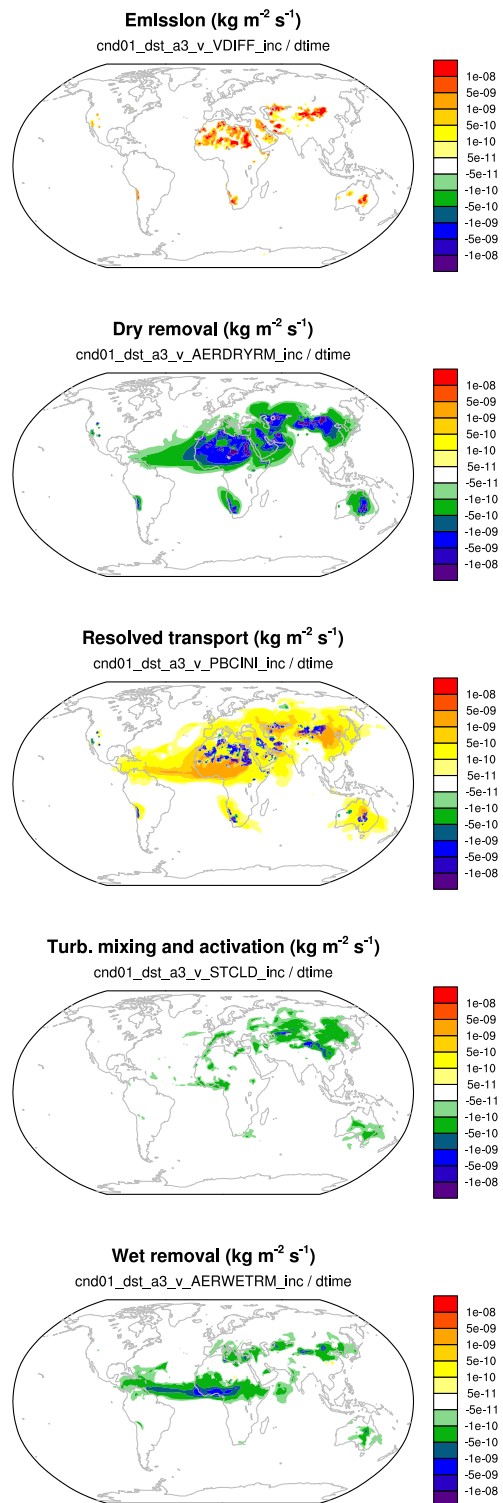

**Figure 5.** One-month mean tendencies of the vertically integrated coarse mode dust burden (unit: kg m$^{-2}$ s$^{-1}$) attributed to different physical processes in EAMv1. The expressions given in thin fonts below panel titles indicate how the presented quantities are calculated from the model's output variables.

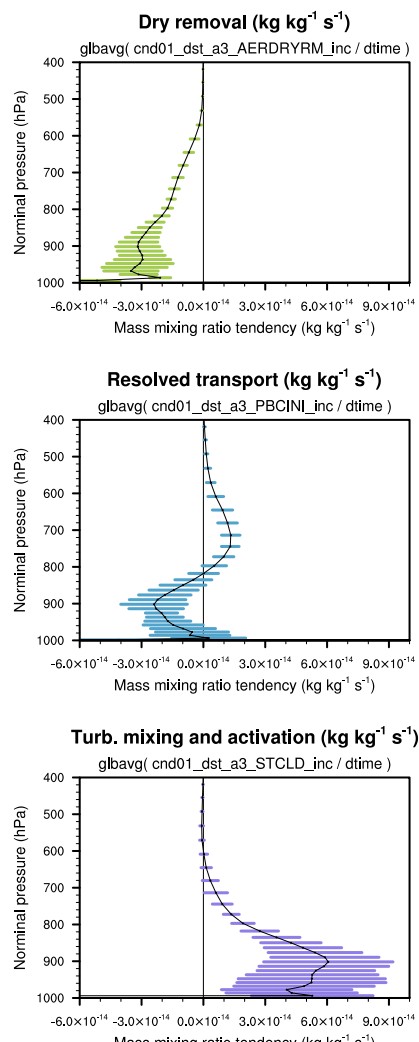

**Figure 6.** Globally averaged vertical profile of the coarse mode dust mass mixing ratio tendencies (unit: kg kg$^{-1}$ s$^{-1}$) attributed to dry removal (top panel), resolved transport (middle panel), as well as the turbulent mixing and activation of aerosol particles (bottom panel). The black curves are monthly averages. The lengths of the horizontal bars correspond to twice of the standard deviation of the daily averages. The expressions given in thin fonts below panel titles indicate how the global averages are calculated from the model's output variables.

**Table 3.** Namelist setup used in the composite analysis presented in Sect. 6.2.

```
metric_name     = 'U10', 'U10', 'U10', 'U10'
metric_nver     = 1,     1,     1,     1
metric_cmpr_type = -1,    0,     1,     1
metric_threshold = 5,     7.5,   10,    -1
metric_tolerance = 0,     2.5,   0,     0
cnd_eval_chkpt  = 'CHEM','CHEM','CHEM','CHEM'

qoi_chkpt = 'CHEM'

qoi_name = 'SFncl_a1', 'SFncl_a2', 'SFncl_a3'
qoi_nver = 1,           1,           1

l_output_state = .true.
l_output_incrm = .false.

hist_tape_with_all_output = 1
nhtfrq = 0
mfilt  = 1
```

### 6.2.1 Simulation setup

In EAMv1, the emission of sea salt aerosol is parameterized with a scheme from Mårtensson et al. (2003) in which the emission flux is proportional to $(U10)^{3.41}$ with U10 being the wind speed (unit: m s$^{-1}$) at 10 m above sea level (Zhang et al., 2016; Liu et al., 2012).

Four conditions are specified in the namelist setup shown in Table 3. The first three divide the possible 10 m wind speed values into three ranges: lower than 5 m s$^{-1}$, between 5 m s$^{-1}$ and 10 m s$^{-1}$, and higher than 10 m s$^{-1}$. The fourth condition

uses the always-fulfilled criterion of U10 > -1 m s$^{-1}$ to select all grid points and time steps for comparison.

Three QoIs are monitored: SFncl_a1, SFncl_a2, SFncl_a3, which are the surface mass fluxes of sea salt aerosol in the accumulation mode, Aitken mode, and coarse mode, respectively. These variable names are EAMv1's standard tracer flux names.

U10 in EAMv1 is the grid-box average provided by the coupler (checkpoint MCTCPL). The calculation of sea salt emissions

is done in the atmosphere model (checkpoint CHEMEMIS). U10 and the surface fluxes are calculated only once per time step $\Delta t_{\mathrm{CPLmain}}$ and their values remain available as components of the derived-type Fortran variable called cam_in (cf. Table A1). Therefore, as long as we select any checkpoint at or after MCTCPL for assessing U10 combined with any checkpoint at or after CHEMEMIS, and before MCTCPL for monitoring the surface fluxes, the results will be equivalent. In Table 3, the same checkpoint CHEM is used for both namelist parameters cnd_eval_chkpt and qoi_chkpt, as this is the checkpoint right

before the surface fluxes are used to update aerosol tracer mixing ratios.

For output, variables from our tool are included in the h0 file as monthly averages.

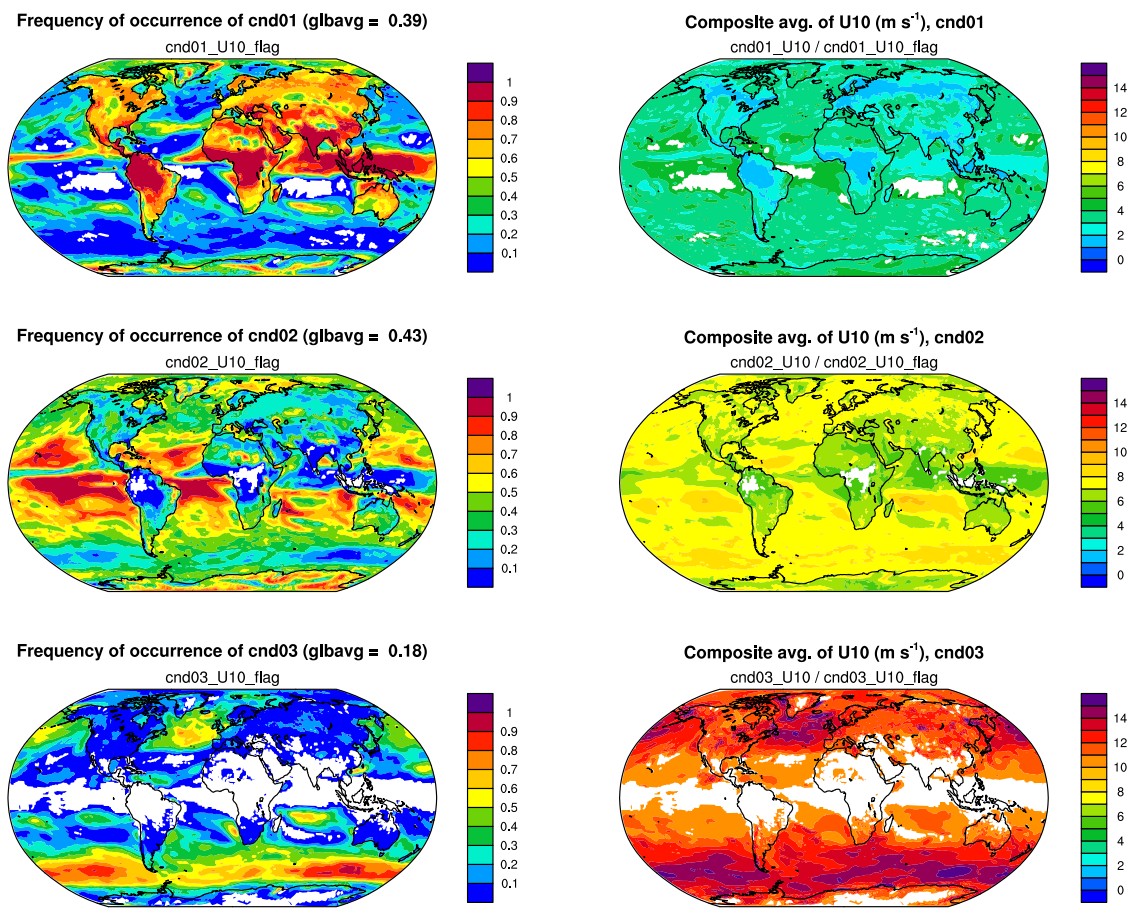

**Figure 7.** Left column: geographical distributions of the frequency of occurrence of conditions 1–3 corresponding to 10 m wind speed (U10) $< 5$ m s$^{-1}$ (top row), between 5 m s$^{-1}$ and 10 m s$^{-1}$ (middle row) and $> 10$ m s$^{-1}$ (bottom row), respectively. Right column: composite average of U10 under each condition. White areas in the contour plots correspond to no occurrence of condition in the one-month simulation. The expressions given in thin fonts below panel titles indicate how the presented quantities are calculated from the model's output variables.

### 6.2.2 Results

Figure 7 presents geographical distributions of the frequency of occurrence of conditions 1–3 in the one-month simulation (left column) and the corresponding composite averages of U10 (right column). While composite averages of U10 are shown for a sanity check, the left panels indicate the different characteristic wind speed associated with different surface types (land versus ocean) and cloud regimes (e.g., deep convection active regions, trade cumulus regions, and storm tracks).

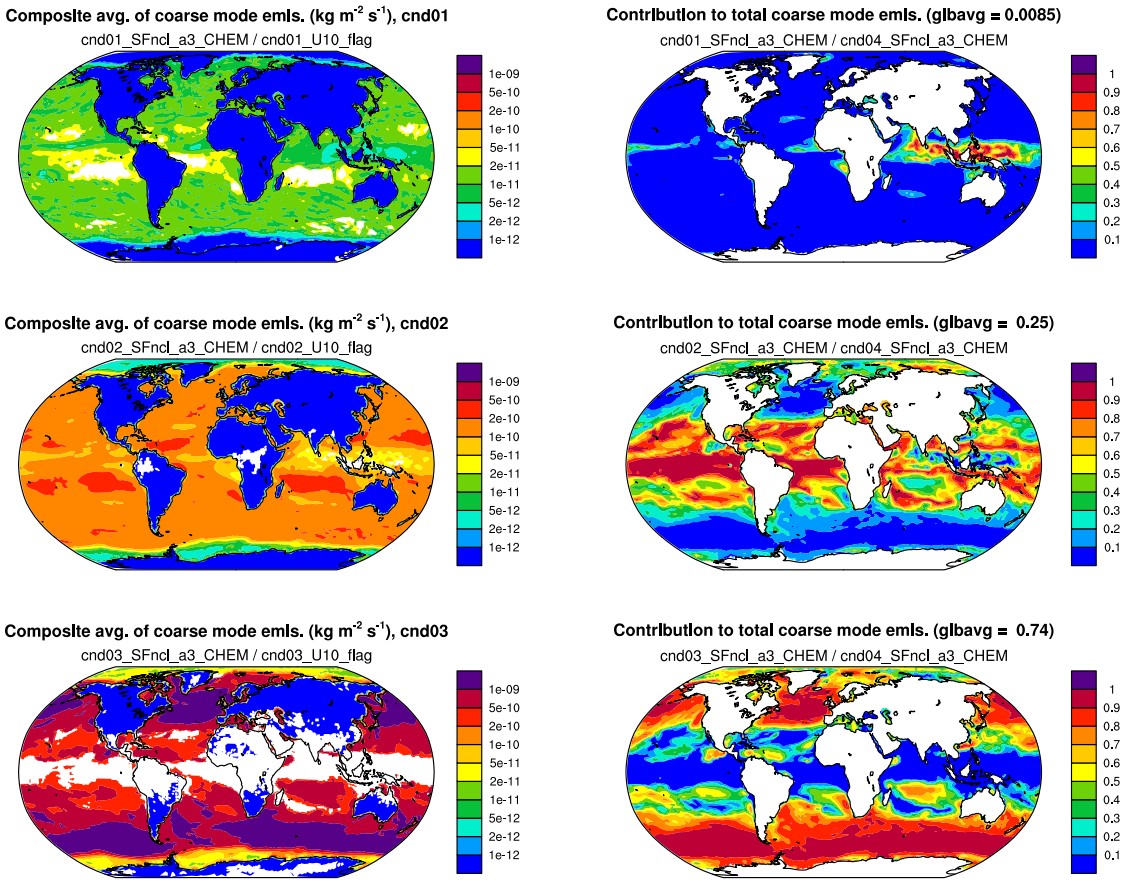

**Figure 8.** Left column: Composite average of coarse mode sea salt mass emission fluxes under conditions 1–3 corresponding to 10 m wind speed $< 5$ m s$^{-1}$ (top row), between 5 m s$^{-1}$ and 10 m s$^{-1}$ (middle row) and $> 10$ m s$^{-1}$ (bottom row), respectively. Right column: contribute of each condition (1, 2, or 3) to the total coarse mode sea salt emission (condition 4). White areas in the left panels are missing values caused by zero frequency of occurrence of the corresponding conditions. White areas in the right panels are missing values caused by zero total coarse mode sea salt emission. The expressions given in thin fonts below panel titles indicate how the presented quantities are calculated from the model's output variables.

Figure 8 shows geographical distributions of the composite mean of the coarse mode sea salt mass emission fluxes under conditions 1–3 (left column) and the relative contribution of each condition to the total (all-condition) fluxes (right column). Here, for demonstration purposes, we only chose three wind speed bins and monitored sea salt mass fluxes. If one refines the

620 wind speed ranges (e.g., use 10 to 20 bins), adds aerosol number fluxes to the QoIs, and adds the calculation of global averages to postprocessing, then diagrams like Fig. 5 in Zhang et al. (2012) can be created to investigate the simulated relationship between wind speed and particle size distribution of the emissions but without having to write out a large amount of instantaneous model output.

## 6.3 A conditional budget analysis for RHI

The third example demonstrates a combined use of the budget analysis and conditional sampling capabilities using our tool. The example also requires the calculation of a diagnosed quantity (the relative humidity with respect to ice, RHI) that is not a state variable, so additional routines are invoked to calculate it. This quantity would vary before and after code compartments (e.g., atmospheric dynamics, cloud microphysics, radiation etc) that operate on the atmospheric state, so it is sensitive to how and where it is calculated in the model, and its value can also change across the sub-cycles used for the parameterizations and their coupling.

**Table 4.** Namelist variables pertaining to the new diagnostic tool used in the conditional RHI budget analysis presented in Sect. 6.3.

```
metric_name      = 'RHI',      'RHI'
metric_nver      =   72,         72
metric_cmpr_type =    1,          1
metric_threshold = 125,         -1
cnd_eval_chkpt   = 'CLDMAC01', 'CLDMAC01'
cnd_end_chkpt    = 'PBCDIAG',  'PBCDIAG'

qoi_chkpt = 'PBCDIAG',  'RAD',      'PACEND',
            'DYNEND',   'DEEPCU',
            'CLDMAC01', 'CLDMIC01',
            'CLDMAC02', 'CLDMIC02',
            'CLDMAC03', 'CLDMIC03',
            'CLDMAC04', 'CLDMIC04',
            'CLDMAC05', 'CLDMIC05',
            'CLDMAC06', 'CLDMIC06'

qoi_name = 'RHI', 'Q', 'QSATI'
qoi_nver =  72,    72,  72

l_output_state = .true.
l_output_incrm = .true.

hist_tape_with_all_output = 1
nhtfrq = 0
mfilt  = 1
```

### 6.3.1 Simulation setup

The focus QoI in this example is the relative humidity with respect to ice (RHI), which directly affects the formation of new ice crystals. In EAMv1, ice nucleation is calculated after the parameterization of turbulence, shallow convection, and large-scale condensation represented by CLUBB (Cloud Layers Unified By Binormals, Golaz et al., 2002; Larson et al., 2002; Larson and Golaz, 2005; Larson, 2017). CLUBB, ice nucleation, droplet nucleation, and other stratiform cloud microphysical

**Table 5.** For the conditional RHI budget example presented in Sect. 6.3: atmospheric processes corresponding to increments diagnosed at the checkpoints selected in Table 4.

| Checkpoint | Atmospheric processes |
|---|---|
| PBCDIAG | Wet removal and resuspension of aerosols |
| RAD | Radiation |
| PACEND | Rayleigh friction and gravity wave drag |
| DYNEND | Resolved dynamics and tracer transport |
| DEEPCU | Deep convection |
| CLDMAC[01–06] | Turbulence and shallow convection, sub-steps 1–6 |
| CLDMIC[01–06] | Stratiform cloud microphysics, sub-steps 1–6 |

processes represented by the parameterization of Gettelman and Morrison (2015) are collectively sub-cycled six times per $\Delta t_{\mathrm{CPLmain}}$. Therefore in the namelist setup shown in Table 4, a checkpoint is selected before each invocation of the ice nucleation parameterization (`CLDMAC01`,..., `CLDMAC06`) to identify sources of high RHI. Additional checkpoints are selected after each invocation of the stratiform cloud microphysics (`CLDMIC01`,..., `CLDMIC06`) to monitor how RHI decreases due to those processes. A few other checkpoints are also selected to evaluate the impact of atmospheric processes that are known to affect air temperature and specific humidity, for example large-scale dynamics, radiation, and deep convection.

In addition to monitoring RHI, we include the specific humidity (Q) and the saturation specific humidity respect to ice (QSATI) as QoIs to help attribute the diagnosed RHI changes (cf. namelist variable `qoi_name` in Table 4). While Q is one of the prognostic variables in EAMv1, RHI and QSATI need to be diagnosed at each checkpoint using three components of the model's prognostic state: Q, air temperature, and pressure. The diagnostic subroutines are included in the module `misc_diagnostics`.

All of the selected QoIs are 3D variables defined in 72 layers in EAMv1. Unlike in the previous example, `qoi_x_dp` is not specified here; it gets the default values of zeros, therefore no vertical integrals are calculated for the QoIs.

Two sampling conditions are specified: the first one selects grid cells where RHI seen by the first invocation of the ice nucleation parameterization is higher than 125%, which is a necessary although insufficient condition to trigger homogeneous ice nucleation. (For clarification, we note that RHI discussed here is the relative humidity calculated from the grid-box mean specific humidity and grid-box mean air temperature. EAMv1 uses RHI > RH$_0$ as a screening condition to determine if homogeneous ice nucleation can occur in a grid box. RH$_0$ depends on air temperature but has typical values around 125%.)

The second condition effectively selects all grid cells and time steps, but we state the condition as RHI > -1% instead of using the special metric "ALL", and select the same condition-evaluation checkpoint as in condition one, so that the conditionally sampled metric `cnd01_RHI` and unconditionally sampled `cnd02_RHI` can be directly and conveniently compared. (Using the special metric "ALL" would result in a metric variable `cnd02_ALL`, which is a constant field of 1.0, being written to the output files.)

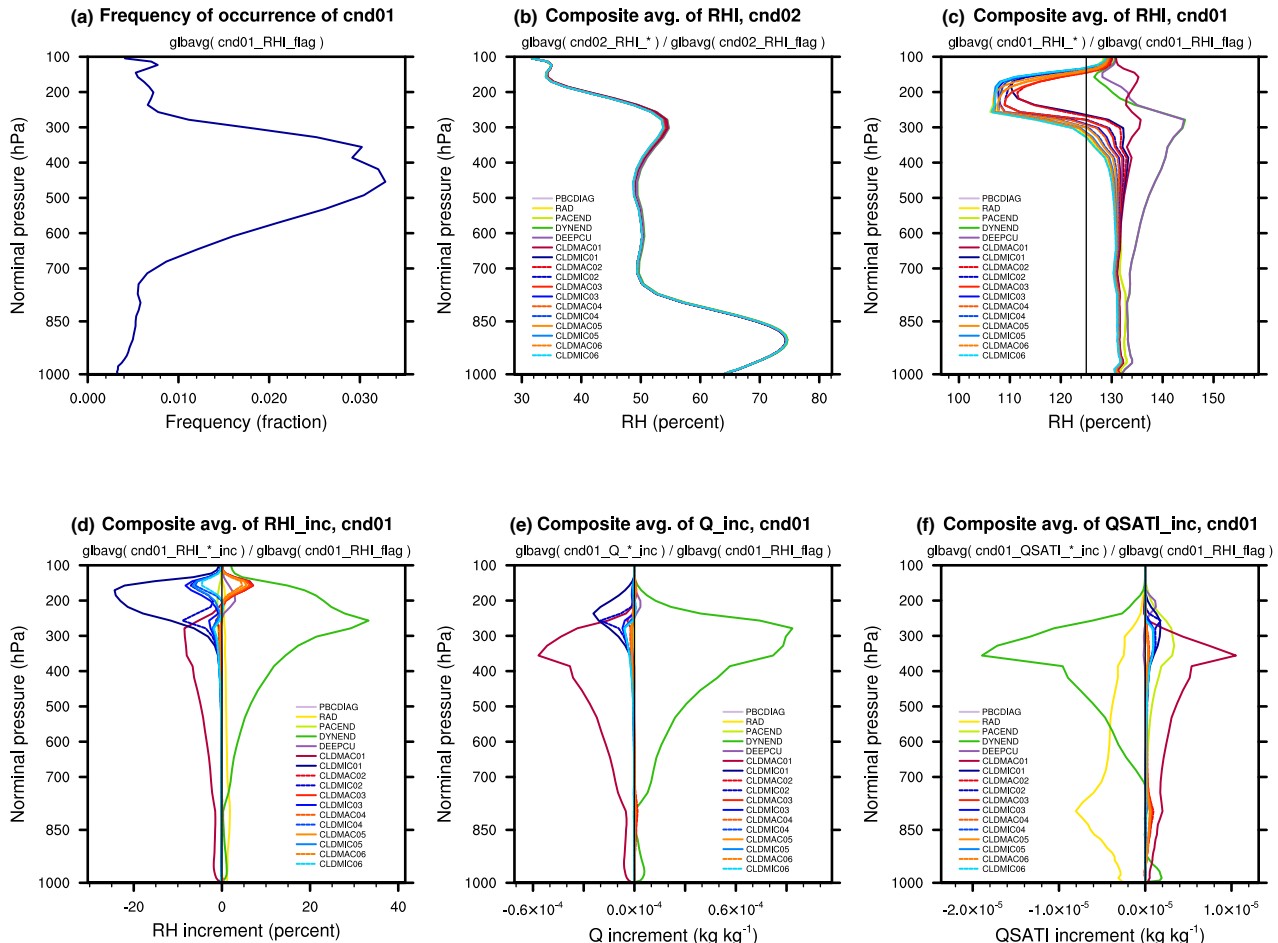

**Figure 9.** Upper row: (a) The frequency of occurrence of RHI > 125% averaged over one month and the entire globe. (b) RHI at various checkpoints averaged over all time steps of the month and over the entire globe (i.e., RHI under condition 02 – unconditional sampling). (c) Space-and-time mean RHI at various checkpoints under condition 01 (i.e., RHI is higher than 125% before the first ice nucleation calculation during a time step of $\Delta t_{\mathrm{CPLmain}} = 30$ min). Lower row: space-and-time mean increments of (d) RHI, (e) specific humidity, and (f) saturation specific humidity with respect ice averaged under condition 01.

The checkpoint before the radiation parameterization is considered the end of a full model time step and hence `cnd_end_chkpt`
is set to `PBCDIAG`. Both the field values and increments of the QoIs are monitored and included in model output. The full set of fields tracked by our tool are sent to output tape 1 (the h0 file) which contains the monthly averages.

### 6.3.2  Results

Figure 9 shows various vertical profiles derived from the simulation. Defining a 2D global average as the average over all grid cells on a sphere weighted by their spherical area, panel (a) in Fig. 9 shows the vertical profile of the 2D global average of the

output variable cnd01_RHI_flag, which gives the globally and temporally averaged frequency of occurrence of RHI > 125% in each grid layer. The other panels in the figure are global averages of different QoIs and checkpoints divided by the global mean frequency of occurrence of the corresponding condition. Recall that our tool assigns a fill value of zero to grid cells and time steps that are unselected for a sampling condition. The profiles in Figures 9b-f are therefore spatial and temporal averages of the corresponding composites.

Panels (b) and (c) show RHI profiles under conditions 02 and 01, respectively. Sampling using the criterion of RHI > 125% helps to highlight the substantial changes related to ice cloud formation in the upper troposphere. Panel (d) shows the increments of RHI at various checkpoints, allowing for a direct comparison of the signs and magnitudes of RHI changes caused by different physical processes. The increments of specific humidity and saturation specific humidity shown in panels (e) and (f) can further help to understand the physical mechanisms causing the RHI changes.

## 7 Conclusions and outlook

An online diagnostic tool has been designed for and implemented in the global atmospheric circulation model EAMv1. The motivation is to introduce a systematic way to support conditional sampling and budget analysis in EAM simulations, so as to (1) minimize the need for tedious ad hoc coding and hence save code development time and avoid clutter, and to (2) reduce the need for instantaneous model output and hence improve the computational efficiency of EAM simulations in which composite or budget analysis is needed.

Building upon the sequential splitting method used by EAM's time integration and the flexibility of the model's output functionalities, the new tool adds its own data structures and functionalities to allow the users to select sampling conditions and model variables (also referred to as quantities of interest, QoIs) to monitor at desired locations of the model's time integration cycles. The condition metrics and QoIs can be any physical quantities that are components of EAM's existing derived-type data structures such as the physics state, physics buffer, and the data structures used for information exchanges between the atmosphere and the surface models such as land and ocean. The condition metrics and QoIs can also be any physical quantities that can be diagnosed from components of these existing data structures. Both the evolving values of the QoIs and their increments caused by different atmospheric processes can be monitored and written out as instantaneous or time-averaged values in EAM's output files (also known as history tapes). For QoIs defined at mid-points of the model's vertical grid or as layer averages, the tool also provides the functionality to calculate and output vertical integrals weighted by the mass of dry or moist air. Multiple sampling conditions can be used in a single simulation. Unconditional sampling and mixtures of conditional and unconditional sampling are also supported.

Assuming the user-chosen conditional metrics and QoIs as well as the locations in time integration cycle to monitor these quantities (referred to as checkpoints) are known to the tool, carrying out a composite or budget analysis using the new tool only requires setting a small number of namelist parameters. The addition of new conditional metrics, QoIs, and checkpoints is straightforward if the data to be sampled can be assessed through EAM's existing data structures.

The new tool has been designed for and implemented in EAMv1 and can be easily ported to EAMv1's descendants (e.g., EAMv2) or predecessors (e.g., CAM5) that use similar Fortran data structures and time integration strategies. Details of the design concepts and implementation in EAMv1 are explained in the paper together with three use case examples that demonstrate the usage of the tool.

The development of the new tool was motivated by the need to carry out conditional budget analysis to understand sources of time-step sensitivities and time-stepping errors related to EAMv1's physics parameterizations. While the current version of the tool, CondiDiag1.0, fulfills the authors' initial needs in those investigations, we are aware of several aspects in which the tool can be further extended or improved to benefit a wider range of EAM users:

First, if the desired condition metric or QoI is calculated by a lower-level (in software sense) subroutine and is not saved in EAM's derived-type data structures (e.g., physics state, physics buffer, etc.), the most convenient way to pass data to CondiDiag will be adding the desired physical quantity to the physics buffer. Such cases will be further assessed and alternative methods will be explored. It is worth noting, however, that the E3SM project has been developing a brand new code base for its version 4 release. The new code uses a single "field manager" for information exchanges between the host model and any resolved or parameterized atmospheric processes. The implementation of our tool in the new code base should make use of – and will benefit from – this new "field manager".

Second, the specification of a sampling condition in CondiDiag1.0 takes the form of a logical expression involving the comparison of a single metric with a threshold value. Section 6.2 demonstrated how the tool can be used for a univariate probability distribution analysis. It will be useful to further extend the tool to support sampling conditions involving multiple metrics and a series of threshold values for each metric, and hence facilitating multivariate probability distribution analysis. Along that line, it might be useful to support sampling conditions involving multiple metrics evaluated at different checkpoints. This could be useful for investigating forcing-response relationships of multiple atmospheric processes and for evaluating the behavior of sub-stepped code compartments.

Third, for simulations that involve multiple sampling conditions, the current tool monitors the same set of QoIs and checkpoints under all conditions. It will be useful to provide the flexibility to select different QoIs and checkpoints for different conditions.

Beyond the three aspects discussed above, there are some desirable extensions of the tool that will require more substantial revisions of the current design. For example, in CondiDiag1.0, the sampling conditions are re-evaluated (and the QoIs are re-sampled) every model time step. We can, however, imagine cases where a user might want to evaluate a condition at some point of a simulation and monitor the evolution of the atmospheric state in the selected grid cells for longer time periods such as a few hours or a few days. Supporting such use cases will require introducing an additional mechanism to specify for how long the evaluated sampling condition is valid. Furthermore, anticipating possible modifications to the sequential splitting of atmospheric processes in EAMv1, in particular possible future adoption of parallel splitting or hybrid methods, it will be useful to explore how the current design of CondiDiag can be extended to accommodate other process coupling methods.

*Code availability.* The EAMv1 code, run scripts, and postprocessing scripts used in this paper can be found on Zenodo at https://zenodo. org/record/5530188. Two versions of the EAM maint-1.0 code are provided: one with CondiDiag1.0 implemented and one without, both of which can be compiled and run on E3SM's supported computer systems. The Zenodo archive also provides a third, smaller tar ball which contains only the source files that were added or revised during the implementation of CondiDiag1.0. A copy of the corresponding EAMv1 files before the implementation is included in this third tar ball to facilitate comparison.

## 735 Appendix A: Candidate metrics and QoIs in CondiDiag1.0

Tables A1–A3 list the currently available physical quantities that can be used as metrics for conditional sampling or be monitored as QoIs.

**Table A1.** Candidate condition metrics and QoIs that are directly copied from EAM's derived-type data structures. "<cnst_name>" refers to tracer names in EAM. "SF<cnst_name>" refers to variables names of tracer surface fluxes in EAM. `pver` and `pverp` are EAM's variable names for the number of vertical layers and vertical interfaces, respectively. In the standard EAMv1, `pver` is 72 and `pverp` is 73. The rightmost column explains the Fortran derived-type variables and their components from which a metric or QoI's values are obtained. More candidate metrics and QoIs can be added following the example shown by the first code snippet in Sect. 4.1.2.

| Name | Explanation | Vertical dimension size | Data source |
|---|---|---|---|
| <cnst_name> | Advected tracers | `pver` | `state%q` |
| T | Air temperature | `pver` | `state%t` |
| U | Zonal wind | `pver` | `state%u` |
| V | Meridional wind | `pver` | `state%v` |
| OMEGA | Vertical velocity | `pver` | `state%omega` |
| PMID | Pressure at layer midpoints | `pver` | `state%pmid` |
| PINT | Pressure at layer interfaces | `pverp` | `state%pint` |
| ZM | Geopotential height at layer midpoints | `pver` | `state%zm` |
| ZI | Geopotential height at layer interfaces | `pverp` | `state%zi` |
| PS | Surface pressure | 1 | `state%ps` |
| SF<cnst_name> | Sfc. flux of advected tracers | 1 | `cam_in%cflx` |
| LWUP | Longwave upward radiative flux from the surface | 1 | `cam_in%lwup` |
| LHF | Latent heat flux from the surface | 1 | `cam_in%lhf` |
| SHF | Sensible heat flux from the surface | 1 | `cam_in%shf` |
| WSX | Surface stress (zonal) | 1 | `cam_in%wsx` |
| WSY | Surface stress (meridional) | 1 | `cam_in%wsy` |
| TREF | Ref. height surface air temp | 1 | `cam_in%tref` |
| QREF | Ref. height specific humidity | 1 | `cam_in%qref` |
| U10 | 10-m wind speed | 1 | `cam_in%u10` |
| TS | Surface temperature | 1 | `cam_in%ts` |
| SST | Sea surface temperature | 1 | `cam_in%sst` |
| FLWDS | Downward longwave flux at surface | 1 | `cam_out%flwds` |
| NETSW | Downward shortwave flux at surface | 1 | `cam_out%netsw` |

**Table A2.** Candidate condition metrics and QoIs that are directly copied from EAM's "physics buffer" data structure. `pver` and `pverp` are EAM's variable names for the number of vertical layers and layer interfaces, respectively. In the standard v1 model, `pver` is 72 and `pverp` is 73. More candidate metrics and QoIs can be added following existing examples in subroutine `get_values` in module `conditiona_diag_main`.

| Name | Explanation | Vertical dimension size | Data source |
|------|-------------|-------------------------|-------------|
| PBLH | Planetary boundary layer height | 1 | `pbuf` |
| TKE | Turbulent kinetic energy | `pverp` | `pbuf` |
| UPWP | Turbulent momentum flux, east-west component | `pverp` | `pbuf` |
| VPWP | Turbulent momentum flux, north-south component | `pverp` | `pbuf` |
| AST | Stratiform cloud fraction | `pver` | `pbuf` |
| CLD | Total cloud fraction (stratiform plus convective) | `pver` | `pbuf` |
| DEI | Cloud microphysics: effective radius of cloud ice for radiation | `pver` | `pbuf` |
| DES | Cloud microphysics: effective radius of snow for radiation | `pver` | `pbuf` |
| MU | Cloud microphysics: size distribution shape parameter for radiation | `pver` | `pbuf` |
| LAMBDAC | Cloud microphysics: size distribution shape parameter for radiation | `pver` | `pbuf` |

**Table A3.** Candidate condition metrics and QoIs that are diagnosed from components of EAM's derived type data structures. `pver` and `pverp` are EAM's variable names for the number of vertical layers and vertical interfaces, respectively. In the standard v1 model, `pver` is 72 and `pverp` is 73. "Subroutine name" is the name of the subroutine in module `misc_diagnostics` that calculates the requested quantity. More candidates can be added following the the second code snippet in Sect. 4.1.2.

| Name | Explanation | Vertical dimension size | Subroutine name |
|------|-------------|-------------------------|-----------------|
| QSATW | Saturation specific humidity w.r.t. water | `pver` | `qsat_water` |
| QSATI | Saturation specific humidity w.r.t. ice | `pver` | `qsat_ice` |
| QSSATW | Supersaturation w.r.t. water given as mixing ratio | `pver` | `supersat_q_water` |
| QSSATI | Supersaturation w.r.t. ice given as mixing ratio | `pver` | `supersat_q_ice` |
| RHW | Relative humidity w.r.t. water in percent | `pver` | `relhum_water_percent` |
| RHI | Relative humidity w.r.t. ice in percent | `pver` | `relhum_ice_percent` |
| CAPE | Convective available potential energy | 1 | `compute_cape` |

## Appendix B: Checkpoints in CondiDiag1.0

Tables B1 and B2 list checkpoints currently implemented in EAM's physics driver subroutines `tphysbc` and `tphysac`.
Table B3 lists the checkpoints in the interface subroutine `clubb_tend_cam`.

**Table B1.** Checkpoints in the parameterization suite calculated before coupling with surface models, i.e., in the `tphysbc` subroutine. The order of checkpoints in the table is the same as the actual order of the checkpoints in the code.

| Model calculations after which checkpoint is implemented | Checkpoint name |
|---|---|
| Dynamical core and large-scale transport | DYNEND |
| Mass and energy fixers | PBCINI |
| Dry adiabatic adjustment | DRYADJ |
| Deep convection | DEEPCU |
| Shallow convection (EAMv0 only) | SHCU |
| CARMA cloud microphysics | CARMA |
| Stratiform cloud macrophysics, sub-step xx | CLDMACxx |
| Aerosol activation and mixing, sub-step xx | CLDAERxx |
| Stratiform cloud microphysics, sub-step xx | CLDMICxx |
| Stratiform clouds, all substeps | STCLD |
| Aerosol wet removal and resuspension | AERWETRM |
| Miscellaneous diagnostics and output | PBCDIAG |
| Radiative transfer | RAD |
| Tropopause diagnosis; export state preparation and output | PBCEND |

**Table B2.** Checkpoints in the parameterization suite calculated after coupling with surface models, i.e., in the `tphysac` subroutine. The order of checkpoints in the table is the same as the actual order of the checkpoints in the code.

| Model calculations after which checkpoint is implemented | Checkpoint name |
|---|---|
| Couling to surface models | MCTCPL |
| Emissions of chemical species | CHEMEMIS |
| Tracer mass fixers | PACINI |
| Chemistry and aerosol microphysics | CHEM |
| Obukov length and friction velocity; Application of surface emissions | CFLXAPP |
| Rayleigh friction | RAYLEIGH |
| Aerosol dry deposition | AERDRYRM |
| Gravity wave drag | GWDRAG |
| QBO relaxation and ION drag | IONDRAG |
| Application of nudging | NDG |
| Dry-to-wet mixing ratio conversion | DRYWET |
| Various diagnostics | PACEND |

**Table B3.** Checkpoints implemented in the "clubb_tend_cam" subroutine. The order of checkpoints in the table is the same as the actual order of the checkpoints in the code.

| Model calculations after which checkpoint is implemented | Checkpoint name |
|---|---|
| Ice saturation adjustment, sub-step xx | ICEMACxx |
| CLUBB, sub-step xx | CLUBBxx |
| Convective detrainment, sub-step xx | CUDETxx |
| Miscellaneous diagnostics, sub-step xx | MACDIAGxx |

## Appendix C: Additional notes on vertical integrals

For aerosol and chemical gases which have a dry-to-wet mixing ratio conversion close to the end of `tphysac` (i.e., before the resolved dynamics and transport), a user might want to track the mixing ratios before and after the conversion as well as at some other checkpoints. In order to inform our tool to use the appropriate air mass for the calculation of vertical integral,

an integer array `chkpt_x_dp` is included in the namelist `conditional_diag_nl`. The values of `chkpt_x_dp` need to be specified in relation to `qoi_chkpt`, i.e., one value of `chkpt_x_dp` for each checkpoint. A value of 1 tells our tool the mass of moist air should be used while a value of 2 indicates dry air mass should be used. Any other values assigned to `chkpt_x_dp` will be interpreted as no specification.

The specifications saved in `chkpt_x_dp` are used by our tool when the namelist parameter `qoi_x_dp` introduced in

Sect. 5.2.4 is assigned values larger than 100. If a value of 101 (moist) or 102 (dry) is specified for an element of the array `qoi_x_dp`, then the corresponding air mass will be used for that QoI at all active checkpoints *except* where `chkpt_x_dp` indicates a different specification. For example, let us assume we set `qoi_x_dp` = 102 for the coarse mode dust mass mixing ratio; we choose to monitor checkpoints A, B, and C and set `chkpt_x_dp` = 0,0,1. Then, when our tool calculates the coarse mode dust burden, checkpoints A and B will use the dry air mass as weights (because `chkpt_x_dp` = 0 for these

two checkpoints means no special treatment while $\mod(qoi\_x\_dp,100) = 2$ for the QoI means dry air mass should be used). For checkpoint C, since `qoi_x_dp` = 102 > 100 and since `chkpt_x_dp` = 1 for the checkpoint, the value 1 from `chkpt_x_dp` will take precedence over $\mod(qoi\_x\_dp,100) = 2$, hence the moist air mass will be used. In other words, a value of `qoi_x_dp` larger than 100 means using $\mod(qoi\_x\_dp,100)$ in general but giving `chkpt_x_dp` precedence when the latter is set to non-zero at a checkpoint.

We acknowledge that the description above is likely not easy to comprehend. For most studies *involving vertical integrals of aerosol or chemical gases*, we recommend *not* using the checkpoints "DRYWET" and "PACEND" listed in Table B2. By doing that, the user will get tracer mixing ratios with consistent definitions at all checkpoints, so that only `qoi_x_dp` (with values 0, 1, or 2) is needed for turning on or off vertical integral.

Last but not least, we clarify that mixing ratios of water species do *not* have this conversion problem.

*Author contributions.* HW designed and implemented CondiDiag in EAMv1 with feedback from the coauthors. KZ and HW designed the use case examples. HW carried out the simulations and processed the results. HW wrote the manuscript; all coauthors helped with the revisions.

*Competing interests.* The authors declare no competing interests.

*Acknowledgements.* We thank the two anonymous reviewers and Dr. Sean Santos for their constrictive comments which helped to improve the manuscript. Cecile Hannay at NCAR is thanked for her help with various versions and formats of CAM's documentation. Computing resources for the initial development and testing of the new tool were provided by the National Energy Research Scientific Computing Center (NERSC), a U.S. Department of Energy (DOE) Office of Science User Facility supported by the Office of Science of the U.S. Department of Energy under Contract No. DE-AC02-05CH11231. Simulations shown as use case examples in this paper were carried out using the DOE Biological and Environmental Research (BER) Earth System Modeling program's Compy computing cluster located at Pacific Northwest National Laboratory (PNNL). PNNL is operated by Battelle Memorial Institute for the U.S. Department of Energy under Contract DE-AC05-76RL01830. The EAMv1 code was obtained from the E3SM project sponsored by DOE BER (DOI: 10.11578/E3SM/dc.20180418.36).

*Financial support.* This research was supported by DOE BER via the Scientific Discovery through Advanced Computing (SciDAC) program (grant no. 70276). KZ was supported by DOE BER through the E3SM project (grant no. 65814).

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
