# Peer review of "CondiDiag1.0: A flexible online diagnostic tool for conditional sampling and budget analysis in the E3SM atmosphere model (EAM)"

_Geoscientific Model Development, 2021_

## Author Comment (AC1)

**Reply on RC1**

**Hui Wan (on behalf of all authors)**

We sincerely appreciate the referee's careful review and positive assessment of the manuscript. Below please find our responses to the specific comments.

**Referee comment:** ll. 181 ff.: If I understand this sentence correctly, it states, that some quantities are calculated at discrete points and become inconsistent with the model's prognostic state (at least as long as the variable field is not updated). Maybe the authors could clarify, that the values consistent with the prognostic state evolve within the time step, but the field values are not updated (only at the points of calculation).

**Author response:** Thanks for the suggestion; a clarification has been added.

The comment also motivated us to add another important note. At the end of the paragraph containing the original line 181, we mention that one needs to be cautious when obtaining values of diagnostic quantities for use by our tool. Later in Section 4.1.2 ("Key algorithm module"), we clarify the following:

Assuming the host model has a diagnostic quantity whose values are saved in the physics buffer under the name `ABC`.

- If the user's intention is to understand the host model *code* by tracking where `ABC` in the physics buffer is updated within a model time step, a code block like the following is needed to retrieve values from the physics buffer (which can be consistent or inconsistent with the prognostic state):

```
case('ABC_PBUF')
  idx = pbuf_get_index('ABC')
  call pbuf_get_field( pbuf, idx, ptr2d )
  arrayout(:,:) = ptr2d
```

- If the user's intention is to understand the *model physics* by monitoring the evolution of `ABC`, a code block like the following is needed to recalculate `ABC` from the current model state:

```
case('ABC_EVOL')
  call calculate_abc( state, ..., arrayout )
```

The RHI budget example shown in Section 6.3 belongs to the second category.

**Referee comment:** l. 346: For readers familiar with EAMs structure and calling sequence it might be clear when (in the model's time stepping) the two subroutines tphysbc and tphysac are called, and it is even written in the manuscript (in the descriptions of Table B1 and B2), but it may be worth to mention the point in the calling sequence here (before/after coupling), which also explains the naming.

**Author response:** This is a good point. Even the developers and users of EAM might find these subsections hard to follow. A new figure is added to the revised manuscript to show the sequence of calculations for the four main parts of EAM: dynamical core, atmosphere-land-ocean exchanges, and the "before-coupling" and "after-coupling" subsets of parameterizations. Also illustrated are code snippets that add checkpoints to `tphysbc` and `tphysac`. In response to a comment from referee 2, we also added a subsection titled "Data and code structures" to the "Host model features" section, in which the subroutines `tphysbc` and `tphysac` are introduced.

**Referee comment:** ll. 350 f.: All the implemented checkpoints are listed in Tables B1 and B2. However, at some point later in the manuscript (l̃. 514) I got lost, as it was not clear to me where to find all the checkpoints and their calling sequence. Maybe the authors could clarify. Suggestion: "All checkpoints added to the EAMv1 subroutines tphysbc and tphysac are listed in Tables B1 and B2, respectively. Most of the checkpoints are implemented by inserting code like..."

**Author response:** Sorry for the confusion. We had a single table listing all of the checkpoints in an earlier draft and then split it up into B1–B3, but some sentences in the manuscript were not updated properly. This is addressed in the revised manuscript.

**Referee comment:** l. 539: "model sub-cycling": Is it the sub-cycling in the model's parameterizations?

**Author response:** For clarity, we replaced the last part of the sentence by "and its value can also change across the sub-cycles used for the parameterizations and their coupling".

**Referee comments:**

Technical corrections

l. 111: "5 hypothetical processes label" → "five hypothetical processes labeled"

ll. 180 f.: "sequential method" → "sequential splitting method"

l. 267: Could a link to Zenodo be inserted here?

l. 386: "carry" → "carry out"

l. 431: add ")" after "checkpoint"

l. 529: "3 wind speed" → "three wind speed"

**Author response:** All corrected or addressed in the revised manuscript. Thanks for the careful review.

---

## Author Comment (AC2)

**Reply on RC2**

Hui Wan (on behalf of all authors)

We thank the referee for the very positive assessment and the helpful questions and suggestions. Our responses to the specific comments are listed below.

**Referee comment:** The manuscript indicated that the version 1 of this tool has implemented in EAM with several successful use cases. EAM is the target host model for now, but it is flexible to be used in other AGCMs. Some possible improvement to make the paper/tool more comprehendible for the general audience:

Elaborate on terminology that is specific to EAM/CAM, just to name few: physics state, physics buffer, cam_in, cam_out, tphysbc vs tphysac...

**Author response:** Thanks for the suggestion. A new subsection is added to Section 2, "Host model features", to explain the terms related to EAM/CAM-specific code structure and data structure.

**Referee comment:** The zenodo doi linked two versions of EAMv1 with and without CondiDiags, but it is not straightforward to see the codes of CondiDiags and how it interfaces with EAM. Not sure about the best approach, but I wonder if the authors can supply with an additional package with standard CondiDiags and a template that could be used as an example for external users?

**Author response:** We plan to add to the Zenodo archive a third, very small tar ball which contains only the source files that were either newly added or had to be revised during the implementation of CondiDiag1.0. A copy of the relevant EAMv1 files before the implementation will be included in this third tar ball as well.

**Referee comment:** Is there a general guideline about adopting ConndiDiags with other AGCM as a host model?

**Author response:** Thanks a lot for the interest. A new subsection (4.4, "Portability") is added to the revised manuscript to discuss this.

**Referee comment:** It is not explicitly indicated that if CondiDiags has already be available in EAM code base and readily to be used for EAM developers?

**Author response:** We developed CondiDiag1.0 using branches in E3SM's public GitHub repository at `https://github.com/E3SM-Project/E3SM`. For example, the revised EAMv1 code shared on Zenodo corresponds to a branch named `huiwanpnnl/maint-1.0_cnd_diag1.0rc`. In recent months, some EAM developers and users expressed interests in using the tool, so we ported it to the code versions they were using (which were typically development versions between v1 and v2). Given the positive feedbacks from the colleagues, we plan to discuss with E3SM's code integrators a pathway to get CondiDiag onto the `main` branch of the E3SM repository. Before that merge happens, our development branches are publicly available in the E3SM repository, and we would be happy to help port the tool to other colleagues' working branches.

**Referee comment:** Technical corrections: In session 2.1, the definition of "process" and "component" seem ambiguous. Ln 117-119: indicate that deep convection contains two sub-components, with the parameterization of impact of convection on temperature and humidity, the parameterization of convective momentum transport. Is process B a more frequent case than processes without sub-processes? It might be useful to have some definition on process and component?

**Author response:** Thanks for the feedback and questions. Nomenclature has been a challenge not only for this manuscript but also in our work on numerical process coupling in EAM which motivated the development of CondiDiag. Oftentimes, we use "processes" to refer to physical phenomena and "components" to refer to sections of the source code (for the latter, "compartment" might be a better name). Because the EAM/CAM code is largely modularized by the simulated physics, processes and components (or compartments) often have close relations. In the revised manuscript, we have added a new subsection to describe the data and code structures in EAM. Hopefully, the new text helps to clarify that EAM has a hierarchically modularized code structure in which we often see sub-processes or even sub-sub-processes.

**Referee comment:** Ln 130. Is there a relationship between the location of where outfld is called to check-points?

**Author response:** This is a very insightful question. The locations of `outfld` calls and the checkpoints are related to some extent, as both `outfld` and our `cnd_diag_checkpoint` subroutine have dummy arguments that use data structures dependent on EAM/CAM's column-chunk-based horizontal domain decomposition as well as EAM/CAM's vertical grid. A *single* call of `outfld` is designed to copy (to the history output infrastructure) the values of a *single* physical quantity stored in a *single* integer or floating-point Fortran array. In contrast, `cnd_diag_checkpoint` is designed to obtain values of a (runtime-specified) *collection* of physical quantities using the EAM/CAM-specific *derived-type* variables like `state` and `pbuf` etc. From this perspective, if the values sent to the history output infrastructure are also saved to the derived-type variables, then one could use a single `cnd_diag_checkpoint` to replace many scattered `outfld` calls.

**Referee comment:** Is the location of outfld process-depended?

**Author response:** We are not quite sure what is meant by "process-depended" in this comment. If it means every physical process has its own set of `outfld` calls that are placed inside the parameterization or immediately after a parameterization is invoked, then yes, the location of `outfld` is process-depended.

It is perhaps useful to mention that `outfld` calls are seen before and after many parameterizations and in multiple levels of subroutines. If an `outfld` call copies the values of a physical quantity from an array that is local to the subroutine and these values are not saved to persistent derived-type data structures or passed to other subroutines, then the `outfld` call needs to be placed in the subroutine where the physical quantity is calculated. On the other hand, if the values being written out are saved and hence persist for a while within a time step, then one will have more options for the location of the `outfld` call.

**Referee comment:** For the inactive checkpoint, are they off by default but can be turned on easily with name list change?

**Author response:** All checkpoints are inactive by default. A checkpoint is turned on (i.e., become active) when the user mentions its name in the namelist. Re-compilation of the model source code is not needed for requesting different sets of checkpoints.

**Referee comment:** Ln 237: "If a value of 101 (moist) or 102 (dry) is used, the the...." Shoud be "..., then the"?

**Author response:** Corrected. Thanks.

**Referee comment:** Session 5.2.4 The process of turning on vertical integral and assign moist/dry air mass for mass-weighting is a little hard to comprehend. Maybe try simplifying if possible...

**Author response:** We have heard similar feedback from other colleagues and have rewritten this part. Hopefully the new version is easier to understand.

---

## Author Comment (AC3)

**Reply on CC1**

**Hui Wan (on behalf of all authors)**

Thank you, Sean, for the very insightful comments, questions, and suggestions. As a long-time user of CESM and E3SM, you already provided answers to most of the questions when making your comments. Nevertheless, our responses are provided below.

**Comment:** I find this manuscript to be well-written overall, and to describe a feature that seems very desirable as a long-time user of CESM and E3SM. There are a few comments/questions I had though:

1. I found this behavior described in the paper to be odd:

"If the metric and the QoI are both 3D but have different numbers of vertical layers (e.g., the metric is the air temperature defined at layer midpoints while the QoI is the net longwave radiative flux defined at layer interfaces), then masking will be skipped, meaning this specific QoI will be captured for output as if no conditional sampling had happened."

My first reaction to this was that the code should raise an error in this case rather than proceed without conditional sampling, since it's surely a user error if a QoI is listed that is incompatible with the metric dimensions. It's only later that the manuscript mentions that the QoIs have to be the same for all conditions, i.e. there may be multiple conditions with different metrics and the list of QoIs cannot be tailored to each condition separately. I assume that this is the reason why the code has to allow a QoI to be specified in this way rather than raising an error, but when reading the paper in order, it's not clear why this behavior was chosen.

**Response:** You are very right that this oddity resulted from the simpler design we chose for version 1 of CondiDiag, namely there may be multiple conditions with different metrics of different dimensions and the list of QoIs cannot yet be tailored to each condition separately. This explanation is added to the revised manuscript at the place you quoted. Thank you.

In the conclusions section of the original manuscript, we said "the current tool monitors the same set of QoIs and checkpoints under all conditions ... we will assess the trade-off between more flexibility and the potential risk of causing confusion for model developers and users." In the past few months, we have already seen multiple new use cases in our own numerics work that this simpler design can lead to a large number of unnessary variables in the history file. So, we probably will change this in version 2 of CondiDiag.

**Comment:** 2. In order to better understand how portable the code is, is it true that the conditional_diag module does not use EAM-specific data structures, but that the other three modules do to some extent?

**Response:** I would say all four modules use some EAM-specific data structures and software functionalities. Some of them would be straightforward to port and some would need a rewrite:

- The `conditional_diag` module has the weakest dependency on EAM/CAM: its meta-data handling part (i.e., parsing the user's choices of QoIs, metrics, etc.) is independent of EAM's data structures. The module also (at least currently) contains a few subroutines that allocate memory for the derived-type arrays used for storing the QoIs, metrics, etc., which assume a column-chunk structure for the horizontal grid.

- The `conditional_diag_main` module assumes all QoIs and condition metrics can be retrieved or re-calculated from EAM/CAM-specific data structures like `state`, `pbuf`, `cam_in` and `cam_out`, so this

module cannot be used in other GCMs without adaptation. On the other hand, I would say the more important part of the module is the general algorithms used for retrieving field values, deriving increments, calculating vertical integrals, and performing conditional sampling, etc. If we understand how another GCM's "state" and "pbuf" etc. are structured, the adaptation is likely to be relatively straightforward.

- The `conditional_diag_output_utils` module uses EAM/CAM's `addfld` and `add_default` subroutines which define variables/fields in the history output files. The part that handles history variable names to distinguish different sampling conditions, checkpoints, etc., can be simply taken to another GCM.

- The `conditional_diag_restart` module is completely EAM/CAM-specific as the subroutines therein read and write restart files. When porting to another AGCM, this module would need to be rewritten following how restart-related I/O is done in the new host model.

**Comment:** Also, since the vertical coordinate must be known in order to perform averaging, is the vertical dimension always the last array dimension in the code?

**Response:** I suppose you mean the vertical integral? In the tiny subroutine `mass_wtd_vert_intg` which calculates the integral, it is assumed that the vertical dimension is the second dimension of the input array (the first dimension is assumed to be column). But I think this can be easily adapted, as Fortran's `sum` function has an optional input argument for specifying which dimension to sum over. For EAM, we currently have

```
! Vertical sum divided by gravit

arrayout(1:ncol,1) = sum( tmp(1:ncol,:), 2 )/gravit
```

If a new host GCM uses, say, rank-3 arrays with dimensions (z, y, x), then we could change the code snippet to the following:

```
! Vertical sum divided by gravit

arrayout(1,1:ny,1:nx) = sum( tmp(:,1:nx,1:ny), 1 )/gravit
```

**Comment:** 3. For Tables B1 and B2, it may be good to mention that these order of checkpoints in the table is the same as the actual order of the checkpoints in the code (assuming that this is the case).

**Response:** This is indeed the case, and the comment is added to the table captions in the revised manuscript. Thanks for the suggestion.

---

## Author Response (AR1)

**Point-by-point response to the reviews**

Hui Wan (on behalf of all authors)

March 2022

We sincerely appreciate the careful reviews and constructive comments from both referees and from Dr. Sean Santos who provided community comments. Considering their suggestions, the following changes have been made to the manuscript:

1. A new subsection titled "Data and code structures" has been added to Section 2 ("Host model features"), in which various EAM-specific terms like the subroutines `tphysbc` and `tphysac` and the derived-type data structures like physics state, physics buffer, import and export state, etc. are explained. A new schematic is added and shown as Fig. 1 to illustrate these terms and their relationships.

2. The concepts of active and inactive checkpoints are clarified in Section 3.1.

3. The possible inconsistencies between the values of diagnostic variables and the values of prognostic variables are clarified in Section 3.2.1. The implications are discussed in Section 4.1.2.

4. A new subsection (4.4, "Portability") has been added to discuss what code changes are needed to port CondiDiag to a new host model.

5. Section 5.2.4, "Turning on vertical integral", has been simplified. The harder-to-comprehend contents have been revised and moved to Appendix C.

6. The use of the words "process" and "component" was reviewed. We have clarified that "atmospheric processes" refers to phenomenon occurring in the atmosphere. The term "model component" is replaced by "code compartment", with the latter referring to subroutines as well as code blocks of numerical treatments, etc.

7. Typo corrections and minor wording revisions have been made throughout the manuscript.

8. The Zenodo archive has been updated. A smaller tar ball has been added that contains only the CondiDiag-specific modules and the revised EAM files (as well as the original version, for comparison).

Our point-by-point responses to the reviewers' comments are given below.
* * *
**Referee 1**
* * *
**Referee comment:** ll. 181 ff.: If I understand this sentence correctly, it states, that some quantities are calculated at discrete points and become inconsistent with the model's prognostic state (at least as long as the variable field is not updated). Maybe the authors could clarify, that the values consistent with the prognostic state evolve within the time step, but the field values are not updated (only at the points of calculation).

**Author response:** Thanks for the suggestion. We rewrote the sentence as follows:

"For diagnostic quantities (e.g., relative humidity), the values consistent with the prognostic state also evolve within each time step even though the arrays in the programming language can temporarily contain inconsistent values until the next time of calculation."

The comment also motivated us to add another important note. At the end of the paragraph containing the original line 181, we mention that one needs to be cautious when obtaining values of diagnostic quantities for use by our tool. Later in Section 4.1.2 ("Key algorithm module"), we clarify the following:

"Here, it is worth pointing out one important caveat for obtaining values of diagnostic quantities in the host model. As mentioned in Sect. 2.1, the values of diagnostic quantities that are consistent with the prognostic state effectively evolves within a full model time step but the arrays in the programing language might have only one or a few updates per full time step and hence can temporarily have inconsistent values. Care is needed to handle the corresponding code blocks in subroutine `get_values` of module `conditional_diag_main`. Let us assume the host model has a diagnostic quantity whose value is saved in the physics buffer under the name `ABC`.

If the user's intention is to *understand the host model's code* by tracking *when* the physics buffer's component `ABC` is updated within a full model time step, a code block like the following is needed:

```
case('ABC'//'_PBUF')
  idx = pbuf_get_index('ABC')
  call pbuf_get_field( pbuf, idx, ptr2d )
  arrayout(:,:) = ptr2d
```

If the user's intention is to *understand the physics* by monitoring the values of `ABC` that are consistent with the evolving prognostic state, a code block like the following is needed which recalculates the value of `ABC` from the state variable:

```
case('ABC'//'_EVOL')
  call calculate_abc( state, ..., arrayout )
```

The RHI budget example shown in Sect. 6.3 falls into the second category."

**Referee comment:** l. 346: For readers familiar with EAM's structure and calling sequence it might be clear when (in the model's time stepping) the two subroutines tphysbc and tphysac are called, and it is even written in the manuscript (in the descriptions of Table B1 and B2), but it may be worth to mention the point in the calling sequence here (before/after coupling), which also explains the naming.

**Author response:** This is a very good point. Even the developers and users of EAM might find these subsections hard to follow. A new schematic is added and shown as Fig. 1 in the revised manuscript to show the sequence of calculations for the four main parts of EAM: dynamical core, the coupler (i.e., atmosphere-land-ocean exchanges), and the "before-coupling" and "after-coupling" subsets of parameterizations. In response to a comment from referee 2, we also added a new subsection titled "Data and code structures" to the "Host model features" section, in which the subroutines `tphysbc` and `tphysac` are introduced and the EAM-specific data structures like the physics state and physics buffer etc. are explained.

**Referee comment:** ll. 350 f.: All the implemented checkpoints are listed in Tables B1 and B2. However, at some point later in the manuscript (∼l. 514) I got lost, as it was not clear to me where to find all the checkpoints and their calling sequence. Maybe the authors could clarify. Suggestion: "All checkpoints added to the EAMv1 subroutines tphysbc and tphysac are listed in Tables B1 and B2, respectively. Most of the checkpoints are implemented by inserting code like..."

**Author response:** Sorry for the confusion. We had a single table listing all of the checkpoints in an earlier draft and then split it up into B1–B3, but some sentences in the manuscript were not updated properly. This is addressed in the revised manuscript.

**Referee comment:** l. 539: "model sub-cycling": Is it the sub-cycling in the model's parameterizations?

**Author response:** For clarity, we replaced the last part of the sentence by "and its value can also change across the sub-cycles used for the parameterizations and their coupling".

**Referee comments:**

Technical corrections

l. 111: "5 hypothetical processes label" → "five hypothetical processes labeled"

ll. 180 f.: "sequential method" → "sequential splitting method"

l. 267: Could a link to Zenodo be inserted here?

l. 386: "carry" → "carry out"

l. 431: add ")" after "checkpoint"

l. 529: "3 wind speed" → "three wind speed"

**Author response:** All corrected or addressed in the revised manuscript. Thanks for the careful review.
* * *
**Referee 2**
* * *
**Referee comment:** The manuscript indicated that the version 1 of this tool has implemented in EAM with several successful use cases. EAM is the target host model for now, but it is flexible to be used in other AGCMs. Some possible improvement to make the paper/tool more comprehendible for the general audience:

Elaborate on terminology that is specific to EAM/CAM, just to name few: physics state, physics buffer, cam_in, cam_out, tphysbc vs tphysac...

**Author response:** Thanks for the suggestion. A new subsection titled "Code and data structures" is added to Section 2 ("Host model features") to explain the terms related to EAM/CAM-specific code structure and data structure. We also added a new schematic, now shown as Fig. 1, to the revised manuscript to illustrate the concept of the "before-coupling" and "after-coupling" parameterization groups which corresponds to the subroutines `tphysbc` and `tphysac`.

**Referee comment:** The zenodo doi linked two versions of EAMv1 with and without CondiDiags, but it is not straightforward to see the codes of CondiDiags and how it interfaces with EAM. Not sure about the best approach, but I wonder if the authors can supply with an additional package with standard CondiDiags and a template that could be used as an example for external users?

**Author response:** We have added to the Zenodo archive a third, small tar ball containing only the source files that were added or revised during the implementation of CondiDiag1.0. A copy of the relevant EAMv1 files before the implementation is included in this third tar ball for comparison.

**Referee comment:** Is there a general guideline about adopting ConndiDiags with other AGCM as a host model?

**Author response:** Thanks a lot for the interest. A new subsection (4.4, "Portability") is added to the revised manuscript to discuss this.

**Referee comment:** It is not explicitly indicated that if CondiDiags has already be available in EAM code base and readily to be used for EAM developers?

**Author response:** We developed CondiDiag1.0 using branches in E3SM's public GitHub repository at `https://github.com/E3SM-Project/E3SM`. For example, the revised EAMv1 code shared on Zenodo corresponds to a branch named `huiwanpnnl/maint-1.0_cnd_diag1.0rc`. In recent months, some EAM developers and users expressed interests in using the tool, so we ported it to the code versions they were using (which were typically development versions between v1 and v2). Given the positive feedbacks from the colleagues, we plan to discuss with E3SM's code integrators a pathway to get CondiDiag onto the `main` branch of the E3SM repository. Before that merge happens, our development branches are publicly available in the E3SM

repository, and we would be happy to help port the tool to other colleagues' working branches.

**Referee comment:** Technical corrections: In session 2.1, the definition of "process" and "component" seem ambiguous. Ln 117-119: indicate that deep convection contains two sub-components, with the parameterization of impact of convection on temperature and humidity, the parameterization of convective momentum transport. Is process B a more frequent case than processes without sub-processes? It might be useful to have some definition on process and component?

**Author response:** Thanks for the feedback and questions. Nomenclature has been a challenge not only for this manuscript but also in our work on numerical process coupling in EAM which motivated the development of CondiDiag. Oftentimes, we use "processes" to refer to physical phenomena, and this is clarified in the revised manuscript. In our day-to-day communications so far and also in the original manuscript, "component" is used to refer to a section of the source code. In the revised manuscript, we have replaced "model component" by "code compartment", both in the text and in all schematics.

Because the EAM/CAM code is largely modularized by the simulated physics, atmospheric processes and code compartments often have close relations. In the revised manuscript, we have added two paragraphs in the new subsection 2.1 ("Data and code structures") to help clarify that EAM has a hierarchically modularized code structure in which one can find sub-processes or even sub-sub-processes which corresponding to multiple levels of subroutines (code compartments).

**Referee comment:** Ln 130. Is there a relationship between the location of where outfld is called to check-points?

**Author response:** A single call of `outfld` is designed to copy (to the history output infrastructure) the values of a single physical quantity. Hence, EAM has a large number `outfld` calls scattered in many subroutines at different levels of the subroutine hierarchy. In contrast, `cnd_diag_checkpoint` is designed to obtain values of a collection of physical quantities. Therefore, a single `cnd_diag_checkpoint` call can replace many scattered `outfld` calls.

**Referee comment:** Is the location of outfld process-depended?

**Author response:** We are not quite sure what is meant by "process-depended" in this comment. If it means every physical process has its own set of `outfld` calls that are placed inside the parameterization or immediately after a parameterization is invoked, then yes, the location of `outfld` is process-depended.

It is perhaps useful to mention that `outfld` calls are seen before and after many parameterizations and in multiple levels of subroutines. If an `outfld` call copies the values of a physical quantity from an array that is local to the subroutine and these values are not saved to persistent derived-type data structures or passed to other subroutines, then the `outfld` call needs to be placed in the subroutine where the physical quantity is calculated. On the other hand, if the values being written out are saved and hence persist for a while within a time step, then one will have more options for the location of the `outfld` call.

**Referee comment:** For the inactive checkpoint, are they off by default but can be turned on easily with name list change?

**Author response:** In the revised manuscript, we clarify in the second bullet in Section 3.1 that *"All checkpoints are inactive by default, meaning no information is retrieved, calculated, or archived by our tool. A checkpoint becomes active when the user selects it at run time (via namelist, cf. Sect. 5.2.3)."*

**Referee comment:** Ln 237: "If a value of 101 (moist) or 102 (dry) is used, the the...." Shoud be "..., then the"?

**Author response:** This sentence has been rewritten and moved to Appendix C.

**Referee comment:** Session 5.2.4 The process of turning on vertical integral and assign moist/dry air mass for mass-weighting is a little hard to comprehend. Maybe try simplifying if possible...

**Author response:** Thanks for the feedback. We have heard similar comments from other colleagues, too. In the revised manuscript, we have simplified 5.2.4 to discuss only the namelist parameter `qoi_x_dp`. The nuanced and hard-to-comprehend part is revised and moved to Appendix C, where we also include a suggestion to avoid using some of the tricky checkpoints for studies involving vertical integrals of aerosol or chemical gases.)
* * *
**Community comments:**
* * *
**Comment:** I find this manuscript to be well-written overall, and to describe a feature that seems very desirable as a long-time user of CESM and E3SM. There are a few comments/questions I had though:

1. I found this behavior described in the paper to be odd:

"If the metric and the QoI are both 3D but have different numbers of vertical layers (e.g., the metric is the air temperature defined at layer midpoints while the QoI is the net longwave radiative flux defined at layer interfaces), then masking will be skipped, meaning this specific QoI will be captured for output as if no conditional sampling had happened."

My first reaction to this was that the code should raise an error in this case rather than proceed without conditional sampling, since it's surely a user error if a QoI is listed that is incompatible with the metric dimensions. It's only later that the manuscript mentions that the QoIs have to be the same for all conditions, i.e. there may be multiple conditions with different metrics and the list of QoIs cannot be tailored to each condition separately. I assume that this is the reason why the code has to allow a QoI to be specified in this way rather than raising an error, but when reading the paper in order, it's not clear why this behavior was chosen.

**Response:** Indeed, this oddity resulted from the simpler design we chose for version 1 of CondiDiag, namely there may be multiple conditions with different metrics of different dimensions and the list of QoIs cannot yet be tailored to each condition separately. In the revised manuscript, we have moved this set of sentences to the end of Section 3.3 ("Multiple sampling conditions in one simulation") and added the explanation.

In the conclusions section of the original manuscript, we said "the current tool monitors the same set of QoIs and checkpoints under all conditions...we will assess the trade-off between more flexibility and the potential risk of causing confusion for model developers and users." In the past few months, we have already seen multiple new use cases in our own numerics work that this simpler design can lead to a large number of unnessary variables in the history file. So, we probably will change this in version 2 of CondiDiag. The corresponding sentence in the conclusions part of the revised manuscript has been changed to "It will be useful to provide the flexibility to select different QoIs and checkpoints for different conditions."

**Comment:** 2. In order to better understand how portable the code is, is it true that the conditional_diag module does not use EAM-specific data structures, but that the other three modules do to some extent?

**Response:** We have added a new section (4.4, "Portability") to the revised manuscript that explains the following:

"Our new tool was originally developed for and implemented in version 1 of EAM and was then tested in v2 and some in-between versions. The porting turned out to be straightforward as the basic code and data structures in EAM had not changed. To implement CondiDiag in models outside the EAM/CAM model families will require some significant adaptation. Some thoughts are shared here.

We assume the host model has a few high-level driver subroutines (or one driver) that organizes code compartments corresponding to various atmospheric processes. This, to our knowledge, is common in AGCMs.

Our code also makes use of the fact that the drivers use derived data types to organize a large number of

model variables of interest for physics-oriented or numerics-focused studies. These derived data types make our code more flexible and compact, especially for conditional sampling.

For performing budget analysis, our current algorithm assumes the sequential splitting method is used in the host model. For models that use different coupling methods (e.g, parallel splitting or a mixture of methods), it might be possible to obtain the budget terms directly from the tendencies saved in existing model variables.

The four new modules CondiDiag introduces to EAM (cf. Sect. 4.1) all use some EAM-specific data structures and software functionalities. For porting to a new model, some parts of these modules will be straightforward to port and the other parts will need a rewrite.

The `conditional_diag` module has the weakest dependency on EAM. The meta-data handling part (i.e., parsing the user's choices of QoIs, metrics, etc.) is independent of EAM's data structures. The module also contains a few subroutines that allocate memory for the derived-type arrays used for storing the QoIs, metrics, etc.. The code therein assumes a chunk-based domain decomposition, which likely will need to be adapted to the new host's data structure.

The `conditional_diag_main` module contains subroutines for retrieving field values, deriving increments, calculating vertical integrals, and performing conditional sampling, etc. The subroutines assume all QoIs and condition metrics can be retrieved or recalculated from EAM-specific data structures described in Sect. **??**, hence the dummy variables and their usage will need to be adapted for a new host model.

Module `conditional_diag_output_utils` and module `conditional_diag_restart` will each need a rewrite for a new host. The key task of the subroutines therein is to do I/O for all components of the derived type `cnd_diag_t`. We expect that one needs to follow the host model's way of handling I/O for 2D and 3D variables. The rewrite will likely be somewhat tedious but presumably not difficult."

**Comment:** Also, since the vertical coordinate must be known in order to perform averaging, is the vertical dimension always the last array dimension in the code?

**Response:** We suppose this comment refers to the vertical integral. In the subroutine `mass_wtd_vert_intg` which calculates the integral, it is assumed that the vertical dimension is the second dimension of the input array (the first dimension is assumed to be column). This can be easily adapted if needed, as Fortran's `sum` function has an optional input argument for specifying which dimension to sum over. For EAM, we currently have

```
! Vertical sum divided by gravit

arrayout(1:ncol,1) = sum( tmp(1:ncol,:), 2 )/gravit
```

If a new host GCM uses, say, rank-3 arrays with dimensions (z, y, x), then we could change the code snippet to the following:

```
! Vertical sum divided by gravit

arrayout(1,1:ny,1:nx) = sum( tmp(:,1:nx,1:ny), 1 )/gravit
```

**Comment:** 3. For Tables B1 and B2, it may be good to mention that these order of checkpoints in the table is the same as the actual order of the checkpoints in the code (assuming that this is the case).

**Response:** This is indeed the case, and the comment has been added to the table captions in the revised manuscript. Thanks for the suggestion.